# Restoring Calibration for Aligned Large Language Models:
# A Calibration-Aware Fine-Tuning Approach

**Jiancong Xiao** [* 1]   **Bojian Hou** [* 1]   **Zhanliang Wang** [* 1]   **Ruochen Jin** [1]   **Qi Long** [1]   **Weijie J. Su** [1]   **Li Shen** [1]

## Abstract

One of the key technologies for the success of Large Language Models (LLMs) is preference alignment. However, a notable side effect of preference alignment is poor calibration: while the pre-trained models are typically well-calibrated, LLMs tend to become poorly calibrated after alignment with human preferences. In this paper, we investigate why preference alignment affects calibration and how to address this issue. For the first question, we observe that the preference collapse issue in alignment undesirably generalizes to the calibration scenario, causing LLMs to exhibit overconfidence and poor calibration. To address this, we demonstrate the importance of fine-tuning with domain-specific knowledge to alleviate the overconfidence issue. To further analyze whether this affects the model's performance, we categorize models into two regimes: calibratable and non-calibratable, defined by bounds of Expected Calibration Error (ECE). In the calibratable regime, we propose a calibration-aware fine-tuning approach to achieve proper calibration without compromising LLMs' performance. However, as models are further fine-tuned for better performance, they enter the non-calibratable regime. For this case, we develop an EM-algorithm-based ECE regularization for the fine-tuning loss to maintain low calibration error. Extensive experiments validate the effectiveness of the proposed methods.

## 1. Introduction

Large Language Models (LLMs) (OpenAI, 2023; Anthropic, 2024) have emerged as powerful tools for a wide range of natural language processing tasks (Bubeck et al., 2023; Chowdhery et al., 2023; Touvron et al., 2023; Team et al., 2023). These models, built upon the Transformer architecture (Vaswani et al., 2017), have demonstrated remarkable abilities to process and generate human-like text, making them increasingly integral to various applications. A crucial development in making LLMs more reliable and aligned with human values is *preference alignment* techniques, particularly Reinforcement Learning from Human Feedback (RLHF) (Ouyang et al., 2022) and Direct Preference Optimization (DPO) (Rafailov et al., 2023). However, an important side effect of preference alignment is its impact on model calibration—the relationship between a model's predicted probabilities and its actual accuracy. While pre-trained LLMs typically demonstrate good calibration properties, preference-aligned models become poorly calibrated, as initially observed in GPT-4 with RLHF (OpenAI, 2023). Our investigation reveals that this is a universal issue across different models aligned with various alignment methods. An example is shown in Figure 1 (left), where the calibration performance of a model aligned by DPO illustrates a poor calibration performance.

Understanding and addressing this calibration issue is crucial. First, well-calibrated prediction is essential for reliable decision-making in real-world applications, particularly in high-stakes domains such as legal or healthcare analysis (Savage et al., 2025). Second, overconfident models may mislead users about their capabilities and limitations, potentially leading to inappropriate reliance on model outputs.

In this paper, we conduct a systematic investigation into two fundamental questions: (1) Why does preference alignment affect calibration? and (2) How can we effectively restore calibration while maintaining the benefits of alignment? Our analysis reveals that the preference collapse phenomenon (Xiao et al., 2024a; Chakraborty et al., 2024), a known issue in alignment methods where models excessively favor certain responses, undesirably generalizes to multiple-choice questions, a crucial scenario where calibration performance can be conveniently evaluated. This generalization leads to overconfidence and poor calibration, as models tend to collapse their prediction to one option while ignoring others, regardless of the correctness of their chosen option.

*Equal contribution   [1]University of Pennsylvania, PA, USA. Correspondence to: Qi Long <qlong@upenn.edu>, Weijie J. Su <suw@wharton.upenn.edu>, Li Shen <lishen@upenn.edu>.

*Proceedings of the 42^nd International Conference on Machine Learning*, Vancouver, Canada. PMLR 267, 2025. Copyright 2025 by the author(s).

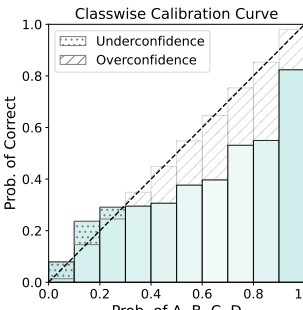 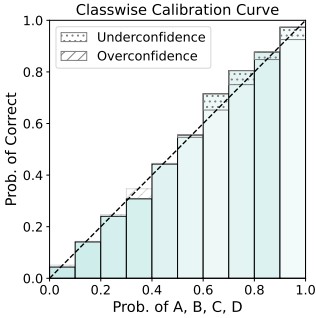 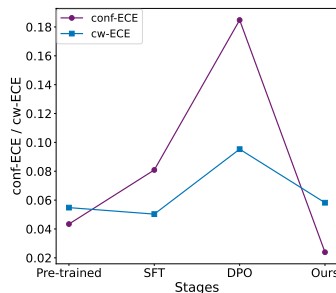

*Figure 1.* Calibration performance comparison between DPO and our approach on Llama3.1-8B-Tulu (a DPO-aligned version of Llama-3.1 (Touvron et al., 2023)). Left: Model calibration plots after DPO alignment, showing significant overconfidence. Middle: Calibration plots after applying our fine-tuning approach, demonstrating improved calibration. Right: The evolution of confidence ECE and classwise ECE across different stages (pre-trained, SFT, DPO, and our method) shows how our approach effectively restores calibration errors.

Building on these insights, we demonstrate that fine-tuning with domain-specific knowledge can extensively alleviate overconfidence on incorrect answers and marginally amplify overconfidence on correct answers, thus improving calibration performance.

For further analysis, we develop a theoretical framework that characterizes the conditions under which calibration can be restored. We introduce the concepts of *calibratable* and *non-calibratable regimes*, defined by upper and lower bounds of Expected Calibration Error (ECE) and a critical accuracy threshold. This framework provides insights into when and how models can achieve good calibration while maintaining performance. We consider fine-tuning the model after RLHF or DPO, rather than modifying these methods directly, as in many practical cases only the final DPO/RLHF model is available, while the pretrained or SFT model is not. We find that preference-aligned models typically lie in the calibratable regime, and propose a calibration-aware fine-tuning (CFT) approach that restores calibration without compromising performance. However, when we overly fine-tune these models to achieve better performance, they shift into the non-calibratable regime. In this case, we identify a fundamental trade-off between ECE and performance, and develop an EM-algorithm-based ECE regularization for the CFT loss to effectively navigate this trade-off.

Through extensive experiments, we demonstrate that our proposed methods significantly improve calibration performance while preserving or even enhancing model's language and knowledge capabilities. Figure 1 (middle) illustrates the calibration plots of our approach and Figure 1 (right) shows the evolution of ECE across different training stages. Our approach reduces the ECE of preference-aligned models from 14.22%–20.10% to 2.39%–6.51% across different model architectures and alignment approaches, which validates the effectiveness of the proposed methods.

## 2. Related Work

**Calibration of Traditional Deep Learning Models.** Prior work has explored various approaches to model calibration. Some researchers have investigated parameterized temperature scaling methods (Guo et al., 2017; Joy et al., 2023; Yu et al., 2022), primarily focusing on vision models. Beyond temperature scaling, researchers have proposed alternative approaches including Dirichlet calibration (Kull et al., 2019), training with label-smoothing (Szegedy et al., 2016), mix-up augmentations (Zhang et al., 2017), and focal loss-based methods (Mukhoti et al., 2020), though these approaches are challenging to implement with LLMs due to their computational demands. Notably, Kumar et al. (2019) conducted rigorous verification of uncertainty calibration, revealing that temperature scaling may be less effective than initially reported.

**Calibration for LLMs.** Several studies have specifically examined LLM calibration (Jiang et al., 2021; Xiao et al., 2022; Chen et al., 2022), identifying miscalibration issues and evaluating various interventions. Some researchers have explored auxiliary models (Zhang et al., 2021; Kadavath et al., 2022; Shen et al., 2024) to improve calibration, while others have investigated prompt-based approaches for RLHF-tuned LLMs (Xiong et al., 2023). Other researchers teach the LLMs to express their uncertainty and calibration (Lin et al., 2022; Tian et al., 2023; Xu et al., 2024b). A distinct line of research focuses on debiasing in-context learning (Abbas et al., 2024; Han et al., 2023; Jiang et al., 2023b), though this differs from statistical calibration. Researchers have studied improving the calibration of LLMs during the SFT phase (Han et al., 2024) as well as during the PPO/DPO phase (Tao et al., 2024; Leng et al., 2024). Notably, previous studies consistently identify temperature scaling as the most effective calibration method. Given its strong performance, we use temperature scaling as our primary baseline for comparison.

# 3. Preliminaries

The calibration error of an LLM is typically evaluated on a multiple-choice question and its corresponding correct answer, which can be formalized as an input-label pair $(x, y)$, where $x$ represents the problem instance consisting of a question stem and four candidate answers, one of which is correct and $y$ is the label of the true answer. While we consider four options in our analysis for concreteness, the framework naturally extends to any number of alternatives. The label $y \in \{A, B, C, D\}$ denotes the position index of the correct answer among the choices.

**Calibration of Classification Models.** Consider a probabilistic classification model $\hat{\boldsymbol{p}} : \mathcal{X} \to \Delta_k$ that outputs class probabilities for $k$ classes $1, \ldots, k$. For any given instance $x \in \mathcal{X}$ it would output some probability vector $\hat{\boldsymbol{p}} = (\hat{p}_1(x), \ldots, \hat{p}_k(x))$ belonging to $\Delta_k = \{(q_1, \ldots, q_k) \in [0,1]^k \mid \sum_{j=1}^{k} q_j = 1\}$ which is the $(k-1)$-dimensional probability simplex over $k$ classes.

In this paper, we mainly consider the classwise calibration (Zadrozny & Elkan, 2002), which requires calibration of all one-vs-rest probability estimators derived from the original multiclass model, and the confidence calibration (Guo et al., 2017), which requires that for instances where the model assigns confidence $c$ to its most probable class prediction, the empirical accuracy should match this confidence $c$.

**Definition 3.1** (Classwise Calibration). A probabilistic classifier $\hat{\boldsymbol{p}} : \mathcal{X} \to \Delta_k$ is classwise-calibrated, if for any class $j$ and any predicted probability $q_j$ for this class:

$$\mathbb{P}(y = j|\hat{p}_j(x) = q_j) = q_j.$$

Classwise-ECE (cw-ECE) is defined as:

$$\text{cw-ECE} = \mathbb{E}_{\hat{\boldsymbol{p}}(x)} \frac{1}{k} \sum_{j=1}^{k} \left| \mathbb{P}(y = j|\hat{p}_j(x)) - \hat{p}_j(x) \right|.$$

**Definition 3.2** (Confidence Calibration). A probabilistic classifier $\hat{\boldsymbol{p}} : \mathcal{X} \to \Delta_k$ is confidence-calibrated, if for any $c \in [1/k, 1]$:

$$\mathbb{P}(y = \text{argmax} \, \hat{\boldsymbol{p}}(x)| \max \hat{\boldsymbol{p}}(x) = c) = c.$$

Confidence-ECE (conf-ECE) is defined as:

$$\mathbb{E}_{\hat{\boldsymbol{p}}(x)} \frac{1}{k} \sum_{j=1}^{k} \left| \mathbb{P}(y = \arg\max \hat{\boldsymbol{p}}(x)| \max \hat{\boldsymbol{p}}(x)) - \max \hat{\boldsymbol{p}}(x) \right|.$$

**Calibration of LLMs.** To answer multiple-choice questions, LLMs typically process the question along with all available answer choices as a single input. The model generates a probability distribution over the possible responses, allowing selection of the most likely correct answer. To ensure

*Table 1.* The format used in the sample in the MedMCQA dataset. The system message is for the Llama family models.

| System: | You are an AI assistant that answers multiple choice questions. You must only respond with a single letter corresponding to your choice without any explanation or additional text. |
|---|---|
| Input: | Which of the following is very difficult to induce antibody? Options: A. Polysaccharide. B. Protein. C. Antigen. D. Effector. Answer with only a single letter: |
| Label: | A |

that LLMs respond with a single letter from $\{A, B, C, D\}$[1], a hint—'Answer with only a single letter'—is appended to the prompt. For models in the Llama family, we use an additional system prompt to enforce this response format. An example of this setup is illustrated in Table 1.

The confidence of LLMs in option $j$ is defined as:

$$\hat{p}_j(x) = \frac{\pi_\theta(y = j|x)}{\sum_{j' \in \{A,B,C,D\}} \pi_\theta(y = j'|x)}. \tag{1}$$

**Preference Optimization.** The RLHF framework for learning LLMs typically consists of the following three steps: (1) supervised fine-tuning (SFT), (2) Reward modeling, and (3) RLHF fine-tuning. The DPO method is to directly optimize the policy without explicitly training the reward function. More details about preference alignment are provided in Appendix A.1.

# 4. A Generative Calibration Framework

## 4.1. A Generative Model for Multiple-Choice Problems

Consider a multiple-choice dataset defined as $\mathcal{S} = \{(x_1, y_1), \ldots, (x_n, y_n)\}$, which can be viewed as an examination created by a test designer. The designer's main constraint is to ensure an approximately balanced distribution, with each option ($A, B, C,$ and $D$) containing about 25% of the correct answers. This test designer can be conceptualized as a probabilistic generative model that assigns correct answers to positions according to their implicit preferences while maintaining this distribution. For a given dataset, there exist many different probabilistic generative models. Here are two illustrative examples:

Example 1: Pure Random Models. The model assigns equal probability to each option regardless of the question: $\mathbb{P}(y = j|x) = 25\%$ for all $j$ and $x$. When generating a dataset $\mathcal{S}$ using this model, we maintain $\mathbb{P}(y = j|\mathcal{S}) = 25\%$.

Example 2: Deterministic Models. For each question $x$,

---

[1]We use the four-option format $\{A, B, C, D\}$ for readability; however, the analysis extends to a general $k$-choice setting.

the model assigns a probability of 1 to exactly one option and 0 to all others: $\mathbb{P}(y = j|x) = 1$ for some $j$ and $\mathbb{P}(y = k|x) = 0$ for all $k \neq j$. The questions are distributed such that approximately 25% of samples assign probability 1 to each option $j$. Using this model, we also maintain $\mathbb{P}(y = j|\mathcal{S}) = 25\%$.

The concept of probabilistic generative models provides a natural connection to calibrated models. Since these models generate the positions of correct answers according to their probability distributions, the observed accuracy always equals the model's confidence (i.e., its predicted probabilities). This inherent property ensures that probabilistic generative models are always well-calibrated. We formalize this relationship in the following proposition.

**Proposition 4.1.** *Let $\boldsymbol{p} : \mathcal{X} \to \Delta_k$ be a probabilistic generative model that assigns correct answers of question $x$ to options $A, B, C$, and $D$ according to distribution $\boldsymbol{p}(x)$. Then $\boldsymbol{p}$ achieves zero classwise and confidence ECE.*

Here, $p(x)$ is unknown. Based on Proposition 4.1, training a well-calibrated LLM can be reformulated as training a probabilistic generative model. However, not all probabilistic generative models are desirable. Consider Example 1, the pure random model: while it achieves zero ECE, it only attains 25% accuracy—rendering it a poorly performing LLM. This illustrates that well-calibration alone is insufficient; we need models that achieve both strong calibration and good accuracy.

**Definition 4.2** (Target probabilistic generative model). Let $\pi_\theta$ be an LLM parameterized by $\theta$, and let $\text{ACC}(\pi_\theta)$ and $\text{ECE}(\pi_\theta)$ denote the accuracy and ECE of $\pi_\theta$ on a dataset $\mathcal{S}$, respectively. Let $\pi^*$ be the optimal solution of

$$\max_\theta \ \text{ACC}(\pi_\theta) \tag{2}$$
$$\text{s.t.} \ \text{ECE}(\pi_\theta) = 0.$$

We define the target probabilistic generative model $\boldsymbol{p}^*$ as the confidence distribution of $\pi^*$ (given by Equation (1)).

Here, $\pi^*$ is not fixed but rather depends on the current state of LLM development, particularly the capabilities enabled by architectural innovations such as the Transformer. As LLM architectures and training methods advance, the achievable accuracy may also improve.

### 4.2. Calibratable and Non-Calibratable Regime

In this subsection, we develop a theoretical framework to analyze when an LLM can achieve perfect calibration by introducing the calibratable and non-calibratable regimes. These regimes are characterized by a critical accuracy threshold by $\pi^*$ that separates them, along with distinct upper and lower bounds for ECE in each regime. To establish the upper and lower bound for ECE, we start from introducing the target calibration error.

**Definition 4.3** (Target Calibration Error (TCE).). Let $\boldsymbol{p}^*$ be the target probabilistic generative model defined in Definition 4.2. The TCE is defined as

$$\text{TCE} = \mathbb{E}_x \frac{1}{k} \sum_{j=1}^{k} \left| p_j^*(x) - \hat{p}_j(x) \right|. \tag{3}$$

To provide better intuition for TCE, we establish its upper and lower bounds. For Theorem 4.4 and 4.5, we assume that the policy $\pi$, can represent an arbitrary probability distribution function over the universe of responses. The region between these bounds is illustrated in Figure 2 (yellow shaded area).

**Theorem 4.4** (Upper Bound of TCE). *Let $\pi^*$ be the target probabilistic generative model. For any accuracy $a \in [0, 1]$, there exists a model $\pi$ with $\text{ACC}(\pi) = a$ such that*

$$\text{TCE}(\pi) \leq 2 \cdot |\text{ACC}(\pi^*) - \text{ACC}(\pi)|. \tag{4}$$

**Theorem 4.5** (Lower Bound of TCE). *Let $\pi^*$ be the target probabilistic generative model. For any accuracy $a \in [0, 1]$ and any model $\pi$ with $\text{ACC}(\pi) = a$, there exists a constant $C \in (0, 1]$, s.t.*

$$\text{TCE}(\pi) \geq C|\text{ACC}(\pi^*) - \text{ACC}(\pi)|. \tag{5}$$

Having introduced the concept of TCE, we can now establish the theoretical bounds for ECE.

**Theorem 4.6** (Upper bound for ECE). *For any model with predicted probabilities $\hat{\boldsymbol{p}}$, the cw-ECE is bounded above by its TCE:*

$$\text{cw-ECE}(\hat{\boldsymbol{p}}) \leq \text{TCE}(\hat{\boldsymbol{p}}).$$

Regarding the lower bound of ECE, we have a trivial bound $\text{ECE} \geq 0$. However, the lower bound has different implications depending on whether a model's accuracy falls above or below the critical accuracy threshold. With this understanding, we can now formally characterize the calibratable and non-calibratable regimes.

**Calibratable Regime.** When a model's accuracy is below the critical accuracy threshold, the following bounds hold for ECE: $0 \leq \text{ECE} \leq \text{TCE} \leq 2(\text{ACC}(\pi^*) - \text{ACC}(\pi))$. We define this region as the calibratable regime, where models can achieve perfect calibration without sacrificing accuracy.

**Non-calibratable Regime.** When a model's accuracy exceeds the critical accuracy threshold, the following bounds hold for ECE: $0 < \text{ECE} \leq \text{TCE} \leq 2(\text{ACC}(\pi) - \text{ACC}(\pi^*))$. We define this region as the non-calibratable regime, where achieving perfect calibration is impossible according to the target probabilistic generative model definition.

The distinction between calibratable and non-calibratable regimes and their corresponding bounds are illustrated in

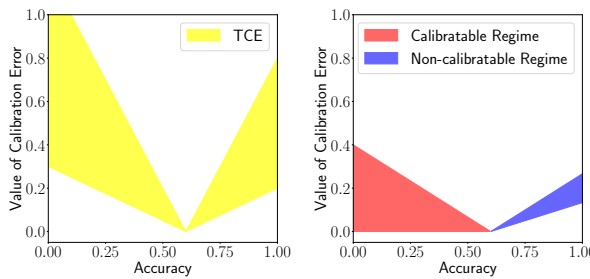

*Figure 2.* Illustration of TCE, calibratable and non-calibratable Regimes. Assume that the target probabilistic generative model $\boldsymbol{p}^*$ has a 60% accuracy. Left: The TCE range (yellow) is bounded between $C|60\% - \text{Accuracy}|$ and $2 \cdot |60\% - \text{Accuracy}|$. Right: The calibratable regime (red) spans from 0 to TCE when accuracy $\leq 60\%$. The non-calibratable regime (blue) spans from a non-zero lower bound to TCE when accuracy > 60%.

Figure 2. What remains undiscussed are the models that lie in the white area above the line of $2 \cdot |60\% - \text{Accuracy}|$. If a model lies in this regime, it has a high ECE, as well as TCE, that is above $2 \cdot |60\% - \text{Accuracy}|$. The model is not well-trained and can be easily fine-tuned to move to one of the calibratable and non-calibratable regimes. In our experiments, we expand the notion of the calibratable regime from accuracy alone to its general performance.

## 5. Algorithms

The objective in Equation (2) can be reformulated as minimizing a combination of accuracy loss and ECE loss using a Lagrange multiplier. In this section, we discuss how to approximate both losses specifically tailored for LLMs.

### 5.1. Approximating the Accuracy Loss for LLMs

In this session, we study why LLMs become poorly calibrated after preference alignment. We first discuss the preference collapse phenomenon (Xiao et al., 2024a; Chakraborty et al., 2024) in alignment methods like RLHF and DPO. When LLMs are aligned with a sample $(x, y_w, y_l)$, they tend to collapse their relative preference on $y_w$ while ignoring $y_l$. This typically results in a preference ratio exceeding human preference proportions: $\pi(y_w|x)/(\pi(y_w|x) + \pi(y_l|x)) > \mathbb{P}(y_w \succ y_l|x)$. This preference collapse issue generalizes to unseen preference pairs $(y_1, y_2)$, where LLMs collapse their preference to one option.

In multiple-choice problems, this collapse manifests as LLMs strongly preferring one option among A,B,C,D, regardless of its correctness. This leads to overconfidence in incorrect answers across many samples, resulting in poor calibration, as demonstrated in Section 6. To address this issue, we propose fine-tuning the model on domain-specific question-answering datasets. We formulate this as an SFT

---

**Algorithm 1** (Regularized) Calibration-Aware FT

**Require:** Number of epochs $L$, Number of bins $M$;
Initialize model $\pi_0$ by the aligned LLMs;
**for** $l = 0$ *to* $L$ **do**
    **E-Step:** // *Use max confidence to stratify samples*
    **for** $i = 1 : n$ **do**
        **for** $m = 1 : M$ **do**
            **if** $\max \text{conf}_{\pi_l}(x_i) \in (\frac{m-1}{M}, \frac{m}{M}]$; **then**
            | $z_i = m$; // $z_i$ *is defined as the latent variable*
            **end**
        **end**
    **end**
    **M-Step:** // *Calibrate model towards accuracy*
    **for** $m = 1 : M$ **do**
        $\mathcal{S}_m = \{(x_i, y_i)|z_i = m, i = 1, \ldots, n\}$;
        $q_m = \frac{1}{|\mathcal{S}_m|} \sum_{(x,y) \in \mathcal{S}_m} \mathbb{1}(\arg\max \text{conf}_{\pi_l}(x) = y)$;
    **end**
    Update $\boldsymbol{p}(x_i)$ by Equation (6), $i = 1, \ldots, n$;
    $\pi_{l+1} = \frac{1}{n} \sum_{i=1}^{n} \min_\pi [\mathcal{L}_{\text{SFT}} + \lambda \mathcal{L}_{\text{ECE}}(\boldsymbol{p}(x_i), \pi_l(x_i))]$;
**end**

---

loss:
$$\mathcal{L}_{\text{SFT}_1} = -\log \pi(y_i|x_i).$$

Through this fine-tuning, models learn domain-specific knowledge. For instance, when trained on the example in Table 1, the model learns that polysaccharides are very difficult to induce antibody production. To further strengthen the model's domain knowledge, we also consider the following loss that focuses more on comprehending the question $x$:

$$\mathcal{L}_{\text{SFT}_2} = -\left[ \log \pi(y|x) + \sum_{t=2}^{T} \log \pi(x^t|x^{t-1}, \ldots, x^1) \right],$$

where $x^t$ are the tokens of question $x$. Here, $\mathcal{L}_{\text{SFT}_1}$ or $\mathcal{L}_{\text{SFT}_2}$ play a role as $\text{ACC}(\cdot)$ in Equation (2) in the LLMs scenario.

### 5.2. Approximating the ECE loss for LLMs

Now, we move our attention to the constraint ECE $= 0$ in Equation (2). Let the confidence of a model $\pi$ be defined as $\text{conf}_\pi(x) = (\hat{p}_A(x), \hat{p}_B(x), \hat{p}_C(x), \hat{p}_D(x))$, where $\hat{p}_j(x)$ is defined in Equation (1). From a probabilistic generative model perspective, the ECE loss can be written as:

$$\mathcal{L}_{\text{ECE}} = \text{D}(\boldsymbol{p}(x), \text{conf}_\pi(x)),$$

where $\boldsymbol{p}(x)$ is some unknown probabilistic generative model and D is the divergence between two distributions. In our experiments, we use the Mean Square Error (MSE) loss. We use an Expectation-Maximization (EM) algorithm to estimate $\boldsymbol{p}(x)$.

For all $\boldsymbol{q} \in \Delta^k$, data points with $\boldsymbol{p}(x) = \boldsymbol{q}$ form a subset sharing the same label generation distribution. Since examining all values of $\boldsymbol{q} \in \Delta^k$ would be impractical, we make

the following simplification. First, for all $q \in \Delta^k$, we define $q = \arg\max q$ and consider the subset $\{x | \max p(x) = q\}$. Second, we discretize $q \in [0,1]$ into $M$ equal-width bins: $[0, 1/M], \ldots, [(M-1)/M, 1]$, consistent with traditional ECE estimation methods. Within the $m$th bin, the generative model assigns answers according to an unknown probability simplex where the largest probability equals $q_m$. Let $I = \arg\max p(x)$. Then, we set $p_{j=I}(x) = q_m$. The probabilities $p_{j \neq I}(x)$ may vary across $x$ within the same bin. We apply a mapping to align these probabilities with the model's confidence scores while maintaining the simplex constraints. We define $p(x)$ as:

$$p_j(x) = \begin{cases} q_m & j = I; \\ \alpha \tanh\left(\gamma \mathrm{conf}_\pi(x)|_j\right) + \beta, & j \neq I, \end{cases} \quad (6)$$

where the value of $\alpha$ and $\beta$ is discussed in Section A.4. The proposed EM-algorithm is provided in Algorithm 1.

# 6. Experiments

## 6.1. Experimental Setup

**Models.** We evaluate our methods using four open-source large language models: Llama-3.1-Tulu-8B (Lambert et al., 2024), Vicuna-7B-v1.5 (Chiang et al., 2023), Olmo 2-7B (OLMo et al., 2024), Mistral-7B (Jiang et al., 2023a), with detailed information provided in Appendix C.1.

We deliberately focus on these four models because each has undergone alignment using either RLHF (Rafailov et al., 2024b) or DPO (Rafailov et al., 2024b) and exhibits notably poor calibration. Specifically, Vicuna-7B is aligned via RLHF, while Mistral-7B, Olmo 2-7B, and Llama-3.1-Tulu-8B are aligned via DPO. Note that the original official Llama-3.1-8B does not provide a clear version among the three-stage pipeline (pre-training, SFT, alignment) and in our tests actually shows decent calibration performance, so we opt to work with the Tulu-8B variant for a more challenging scenario. In this way, we ensure that our calibration methods are thoroughly evaluated against a range of alignment strategies and baseline calibration outcomes.

**Baselines.** We validate the effectiveness of our methods by comparing performance across five methods:

- RLHF or DPO Preference Alignment.
- Temperature Scaling (TS) (Guo et al., 2017): Introduces a temperature parameter $T$ to scale the logits, minimizing ECE on a validation set to achieve better calibration.
- Label Smoothing (Müller et al., 2019): Replaces hard one-hot labels with a smooth label that assigns $1 - \varepsilon$ to the correct class and distributes $\varepsilon$ over the remaining classes.
- Calibration-Aware Fine-Tuning (CFT): In the calibratable regime, we use $\mathcal{L}_{\mathrm{SFT}_1}$ to fine-tune the models.

In this regime, we find that with ($\lambda > 0$) or without ($\lambda = 0$) an explicit ECE regularization produces similar results and we can simply report the unregularized one in our following experiments.

- Regularized CFT (RCFT): We use $\mathcal{L}_{\mathrm{SFT}_2}$ to fine-tune the models. This increases the model accuracy significantly and moves the model to the non-calibratable regime. Therefore, we use an explicit ECE regularization with $\lambda = 1$ in Algorithm 1.

**Evaluation Metrics.** In this paper, we use four metrics to evaluate the effectiveness of the proposed method. They are (1) class-wise Expected Calibration Error (cw-ECE), (2) Conference Expected Calibration Error (conf-ECE) (Guo et al., 2017), (3) Accuracy and (4) Win Rate where the first two are to evaluate the calibration performance (the lower the better). We use Accuracy and Win Rate to evaluate the ability of LLMs to understand languages and answer the questions correctly (the higher the better). The continuous cw-ECE and conf-ECE are defined in Def. 3.1 and 3.2. We use their discrete versions by stratifying all the samples into ten bins according to the prediction probability. The discrete versions can be found in Section C.1.

The Win Rate metric measures how often a model assigns a higher sequence probability to the "preferred" response than to the "non-preferred" one in a pairwise comparison. Specifically, for each pair in the preference dataset, if the model ranks the chosen (preferred) response higher than the rejected one, that counts as a "win," and the Win Rate is the fraction of wins among all comparisons. A more detailed description of Win Rate can be found in Appendix C.1.

We anticipate that the model adjusted using our approach will achieve a competitive Win Rate in comparison to the model prior to calibration. This will demonstrate that the good alignment performance brought by DPO/RLHF will not be diminished when we lift the calibration performance.

**Datasets.** We evaluate the ECE and Accuracy using four diverse datasets: MMLU, MedMCQA, OpenBookQA, Arc-Challenge, and evaluate the Win Rate using AlpacaEval, Arena-Hard, and UltraFeedback-Binarized-Preferences. The detailed descriptions of these datasets are provided in Appendix C.1.

**Evaluation Schemes.** We evaluate not only the *in-domain* performance (i.e., the training and testing data share the same domain), but also the *out-domain* (zero-shot) ability on an out-of-domain dataset. More specifically, we have two schemes:

- **Scheme 1 (In-Domain):** We combine the three datasets MMLU, MedMCQA, OpenBookQA into a single dataset and split it into training (3,000 samples) and testing (2,000 samples) datasets. The model is trained (calibrated) on the training dataset and evalu-

*Table 2.* Performance comparison among DPO/RLHF, Temperature Scaling, Label Smoothing, CFT, and RCFT across four models (**Llama3.1-8B-Tulu**, **Vicuna-7B**, **Olmo2-7B**, and **Mistral-7B**) in in-domain and out-domain scenarios. Best results in each metric block are bold. Blue highlights indicate superior in-domain conf-ECE of our CFT while red highlights denote best in-domain accuracy of our RCFT. "↓"/"↑" means the smaller/larger the better. "-" means the results of Temp. Scale. are the same as the original DPO/RLHF version.

| Model | Method | conf-ECE ↓ | | cw-ECE ↓ | | Accuracy ↑ | |
|---|---|---|---|---|---|---|---|
| | | In-Domain | Out-Domain | In-Domain | Out-Domain | In-Domain | Out-Domain |
| **Llama3.1-8B-Tulu** | DPO | 0.1953 | 0.1212 | 0.0953 | 0.0650 | 0.6228 | 0.7810 |
| | Temp. Scale. | 0.1126 | **0.0679** | **0.0336** | 0.0514 | - | - |
| | Label Smooth. | 0.1898 | 0.1009 | 0.0692 | 0.0639 | 0.6372 | 0.7116 |
| | CFT(Ours) | **0.0239** | 0.0688 | 0.0582 | **0.0375** | 0.6410 | **0.8000** |
| | RCFT(Ours) | 0.0897 | 0.0810 | 0.0771 | 0.0526 | **0.8341** | 0.7991 |
| **Vicuna-7B** | RLHF | 0.1422 | 0.0852 | 0.0979 | 0.0560 | 0.4344 | 0.5233 |
| | Temp. Scale. | 0.0598 | **0.0224** | 0.0488 | **0.0484** | - | - |
| | Label Smooth. | 0.1221 | 0.0823 | 0.0517 | 0.0544 | 0.4517 | 0.5767 |
| | CFT(Ours) | **0.0379** | 0.0331 | 0.0583 | 0.0491 | 0.4481 | 0.6172 |
| | RCFT(Ours) | 0.0474 | 0.0672 | **0.0459** | 0.0530 | **0.6015** | **0.6035** |
| **Olmo2-7B** | DPO | 0.1555 | 0.1325 | 0.0873 | 0.1331 | 0.6210 | 0.6635 |
| | Temp. Scale. | 0.0665 | 0.1160 | **0.0355** | 0.1196 | - | - |
| | Label Smooth. | 0.1010 | 0.0499 | 0.0791 | 0.1298 | 0.6808 | 0.6431 |
| | CFT(Ours) | **0.0544** | **0.0225** | 0.0804 | **0.0637** | 0.6606 | 0.7085 |
| | RCFT(Ours) | 0.0989 | 0.0781 | 0.0806 | 0.0707 | **0.8510** | **0.7099** |
| **Mistral-7B** | DPO | 0.2010 | 0.1318 | 0.0909 | 0.1103 | 0.6331 | 0.7567 |
| | Temp. Scale. | 0.0802 | 0.0991 | **0.0399** | 0.0909 | - | - |
| | Label Smooth. | 0.1874 | 0.1121 | 0.0900 | 0.0990 | 0.6479 | 0.6997 |
| | CFT(Ours) | **0.0651** | **0.0424** | 0.0712 | **0.0614** | 0.6514 | **0.7863** |
| | RCFT(Ours) | 0.0979 | 0.0731 | 0.0877 | 0.0739 | **0.8297** | 0.7768 |

ated on the testing dataset. This setup aims to assess the calibration performance in an in-domain scenario.

- **Scheme 2 (Out-Domain):** The model is calibrated using the same training dataset as the In-Domain approach, which consists of the three datasets: MMLU, MedMCQA, and OpenBookQA, totaling 3,000 samples, and evaluated on the Arc-Challenge dataset (2,000 samples). This scenario tests the model's ability to transfer the calibrated capabilities from one domain to another, highlighting its zero-shot (generalization) potential.

Given our computational resources (four A100 40G GPUs), we employ the Quantized Low Rank (QLoRA) technique (Dettmers et al., 2024) to fine-tune all models with rank = 128, LoRA scaling parameter $\alpha = 64$, and `bfloat16` precision on 2 A100-40G GPUs. The training is conducted for 5 epochs with a batch size of 2 and a learning rate of $5 \times 10^{-6}$.

### 6.2. Numerical Results

Table 2 compares five alignment methods—DPO/RLHF, Temperature Scaling, Label Smoothing, CFT (ours), and RCFT (ours)—across four language models (Llama3.1-8B-

Tulu, Vicuna-7B, Olmo2-7B, and Mistral-7B) in terms of calibration performance (conf-ECE, cw-ECE) and Accuracy under both in-domain and out-domain evaluations.[2] Key insights include:

**CFT significantly improves calibration while preserving or enhancing language capabilities.** For Vicuna-7B, CFT reduces in-domain conf-ECE by 73% compared to DPO (0.0379 vs. 0.1422) while increasing out-domain accuracy to 0.6172 (vs. DPO's 0.5233). Similarly, for Mistral-7B, CFT achieves a 68% reduction in in-domain conf-ECE (0.0651 vs. DPO's 0.201) and maintains strong out-domain accuracy (0.7863 vs. DPO's 0.7567). CFT also outperforms Temperature Scaling and Label Smoothing in challenging scenarios, such as reducing Olmo2-7B's out-domain conf-ECE to 0.0225 (vs. Label Smoothing's 0.0499 and Temperature Scaling's 0.1160). This verifies that DPO or RLHF-aligned models typically lie in the calibratable regime, where calibration can be restored without sacrificing language capabilities.

**RCFT prioritizes accuracy gains while maintaining**

---

[2]Code is publicly available at https://github.com/ZhanliangAaronWang/RestoreLLMCalibration.

*Table 3.* Win rate comparisons among DPO/RLHF (DPO used in Table 3), CFT and RCFT across four models (Llama3.1-8B-Tulu, Vicuna-7B, Olmo2-7B and Mistral-7B) on three datasets (AlpacaEval, Arena-Hard and Ultrafeedback). The best performance for each dataset is in bold. The competitive performance indicates that our methods can preserve the alignment performance.

| Model | AlpacaEval (vs DPO) | | AlpacaEval | | | Arena-Hard | | | Ultrafeedback | | |
| --- | --- | --- | --- | --- | --- | --- | --- | --- | --- | --- | --- |
| | CFT vs DPO | RCFT vs DPO | DPO | CFT | RCFT | DPO | CFT | RCFT | DPO | CFT | RCFT |
| Llama-3.1-8B-Tulu | 51.68 vs 48.32 | 46.83 vs 53.16 | 21.4 | **22.6** | 19.6 | 44.6 | **45.0** | 43.6 | 0.7295 | **0.7460** | 0.7118 |
| Vicuna-7B | 46.46 vs 53.54 | 50.43 vs 49.57 | 2.60 | 2.60 | **3.60** | 1.00 | 1.00 | 1.00 | 0.2271 | **0.2279** | 0.2257 |
| Olmo2-7B | 62.48 vs 37.52 | 46.12 vs 53.88 | **24.2** | 22.9 | 23.1 | 19.4 | 19.2 | **20.2** | 0.7493 | **0.7588** | 0.7517 |
| Mistral-7B | 46.96 vs 53.04 | 49.81 vs 50.19 | 26.0 | **26.8** | 25.2 | **18.9** | 18.3 | 18.0 | 0.7066 | 0.7124 | **0.7221** |

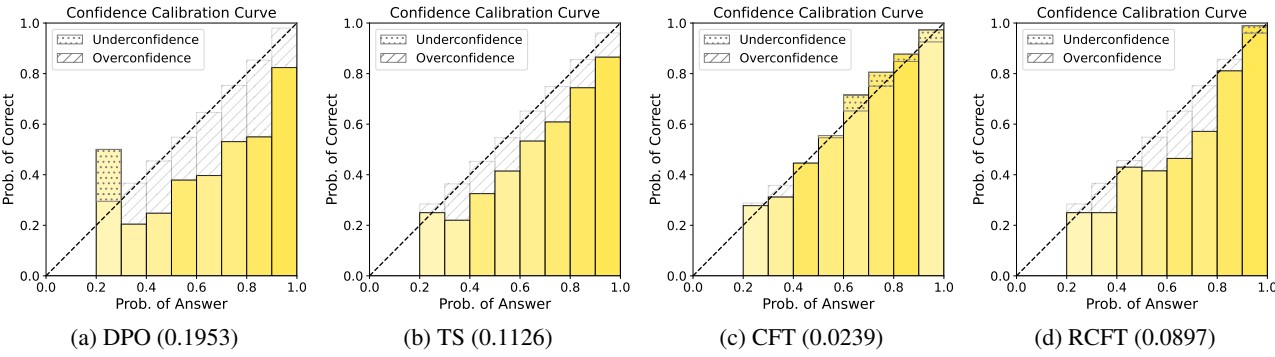

(a) DPO (0.1953)  (b) TS (0.1126)  (c) CFT (0.0239)  (d) RCFT (0.0897)

*Figure 3.* Calibration Plots of (a) DPO, (b) Temperature Scaling (TS), (c) our CFT, (d) our RCFT on Llama-3.1-8B-Tulu. Each panel plots the model's predicted probabilities (i.e., confidence) on the $x$-axis against the observed accuracy (fraction correct) on the $y$-axis, binned into ten groups. The diagonal line in each panel represents *perfect calibration*. The depth of the color indicates the sample density in that column. DPO has the worst calibration performance. Other three methods improve the calibration performance where our CFT has the lowest con-ECE (shown in the parenthesis). This figure omits the first two bins because the model selects an answer with the largest predicted probability which is always larger than 0.25 in the four options prediction task (so no samples exist below that threshold).

**competitive calibration.** For Llama3.1-8B-Tulu, RCFT achieves the highest in-domain accuracy (0.8341) compared to DPO (0.6228) but exhibits slightly higher calibration errors (in-domain conf-ECE: 0.0897 vs. CFT's 0.0239). This trend holds for Olmo2-7B, where RCFT boosts in-domain accuracy to 0.851 (vs. DPO's 0.621 and Label Smoothing's 0.6808) with modest conf-ECE (0.0989). Here, the models move to the non-calibratable regime. While RCFT trades some calibration performance for accuracy, it remains comparable to Temperature Scaling, making it suitable for accuracy-critical applications.

**Baseline methods show consistent limitations.** Temperature Scaling reduces calibration errors (e.g., Vicuna-7B conf-ECE: 0.0598 vs. 0.1422) but cannot improve accuracy (marked by "-"). Label Smoothing provides moderate calibration benefits but underperforms our methods—for example, on Mistral-7B, its in-domain conf-ECE (0.1874) is significantly higher than CFT's (0.0651), and its accuracy gains (+1.5% over DPO) are dwarfed by RCFT's +31%. DPO/RLHF consistently shows the poorest calibration (e.g., Mistral-7B conf-ECE: 0.201), highlighting the need for specialized techniques.

**CFT and RCFT preserve alignment quality across di-**

**verse benchmarks.** Table 3 demonstrates this through three complementary evaluation protocols:

1. **AlpacaEval Head-to-Head (CFT/RCFT vs DPO):** Responses from our methods and the original DPO model were directly compared by GPT-4 (judge):

- *Protocol*: Same prompts → CFT/RCFT vs DPO responses → GPT-4 preference
- *Results*: Statistical parity (e.g., Vicuna-7B: 50.43% RCFT vs 49.57% DPO)

2. **Standard AlpacaEval (vs GPT-4o):** Methods were evaluated against the standard GPT-4o baseline:

- *Protocol*: Responses vs GPT-4o → GPT-4 preference
- *Results*: CFT achieved best performance for Llama3.1 (22.6%) and Mistral-7B (26.8%)

3. **Arena-Hard (vs GPT-4):** Responses were compared against the standard GPT-4 baseline[3]:

- *Protocol*: Responses vs GPT-4 → GPT-4o preference
- *Results*: RCFT achieved best for Olmo2-7B (20.2%);

---

[3]The value of the instruct version of Llama-3.1-8B is a 20.6%. It is not clear why the Tulu version has a significant improvement.

CFT showed gains (Llama3.1: 45.0% vs DPO's 44.6%)

4. **Ultrafeedback Binary Selection:** Models performed preference classification:

- *Protocol*: Given question, response A, response B → Select better response
- *Results*: CFT improved win rates for all models (e.g., Olmo2-7B: 0.7588 vs DPO's 0.7493)

**In summary, CFT excels in calibration-sensitive tasks while RCFT dominates accuracy-critical applications.** Both methods outperform existing techniques (including Label Smoothing), preserve alignment quality across all benchmarks, and demonstrate robust generalization across models and evaluation scenarios.

### 6.3. Calibration Plots

Besides the numerical results, we also illustrate the calibration plots to provide a more intuitive display. Figure 3 shows the confidence calibration plots for the four methods: DPO, Temperature Scaling (TS), CFT, and RCFT on Llama-3.1-8B-Tulu. The horizontal axis of each panel indicates the model's predicted probability (confidence), while the vertical axis is the observed accuracy for predictions within that confidence bin.

- **Perfect calibration:** The diagonal line in each plot indicates *perfect calibration*, meaning a model's predicted confidence aligns exactly with its true probability of correctness. Bars above the diagonal are underconfident; bars below the diagonal are overconfident.
- **(a) DPO (0.1953):** Direct Preference Optimization shows the worst calibration. Its con-ECE value (0.1953) is the highest among the four, and the deviation of its yellow bars from the diagonal line is relatively large, indicating a substantial mismatch between predicted probabilities and actual accuracies.
- **(b) TS (0.1126):** Temperature Scaling reduces overconfidence compared to DPO, bringing the bars closer to the diagonal. Its con-ECE drops to 0.1126, reflecting a moderate improvement in calibration.
- **(c) CFT (0.0239):** This method demonstrates much stronger calibration. The bars closely track the diagonal line, and its con-ECE of 0.0239 is the lowest among the four methods. This indicates that CFT is nearly well-calibrated for the task.
- **(d) RCFT (0.0897):** While RCFT is not as well-calibrated as CFT, it still surpasses DPO and TS in bringing predicted probabilities in line with actual accuracies. Its con-ECE is 0.0897, representing a moderate level of calibration improvement. Note that RCFT has a better accuracy as indicated in Table 2.

In all panels, darker yellow bars indicate higher density of samples within that bin. Since the models' largest confidence $\geq 0.25$ in the four-choice task, the first two bins are effectively empty and thus omitted. Overall, DPO is substantially miscalibrated, while TS, CFT, and RCFT bring the model's confidence closer to true accuracy, with CFT having the best calibration (lowest conf-ECE). Other models share similar phenomenon. We defer other results to Appendix C.3.

To conclude this section, we present an ablation study examining the effects of $\lambda$. Using only $\mathcal{L}_{SFT_2}$ achieves high accuracy ( 90%) but shows poor calibration. Conversely, using only $\mathcal{L}_{ECE}$ drives the model toward random guessing, achieving near-zero ECE but with only 25% accuracy. These results, detailed in Section C.2, demonstrate the importance of balancing both objectives to achieve both good calibration and performance.

## 7. Conclusion

In this work, we have investigated the critical issue of calibration degradation in LLMs following preference alignment procedures. To address this, we introduced a theoretical framework distinguishing between calibratable and non-calibratable regimes, and developed practical solutions through calibration-aware fine-tuning approaches. Our experimental results across multiple models demonstrate that our methods can significantly improve calibration while maintaining or enhancing model performance. Future work could explore extending these methods to other types of language tasks and investigating the relationship between calibration and other aspects of model reliability.

**Limitation and Future Work.** Our findings suggest several promising directions for research on LLM calibration. First, determining which regime a model falls into ultimately reduces to understanding whether its accuracy exceeds a certain threshold. This, in turn, depends on two key factors: 1. What is the accuracy threshold for a given neural network architecture? 2. Given such an architecture, which training algorithms lead the model to fall into each regime? Answering these questions requires a deeper theoretical analysis of the properties of transformers, which is currently an open and challenging direction. Second, from a theoretical perspective, determining the constant $C$ in Theorem 4.5 is nontrivial, and we leave it as an open problem for future work. Third, our current study focuses on multiple-choice settings; extending the method to free-form generation is an important direction for future investigation. Fourth, exploring the effectiveness of our approach in scenarios where the LLM is initially poorly calibrated remains an important direction. Finally, exploring the calibration properties of quantized models, black-box API models, and whether low-rank adaptation methods such as LoRA suffer from calibration issues also presents valuable avenues for future research.

## Impact Statement

Our work advances the reliability and trustworthiness of large language models by improving their calibration while maintaining alignment with human preferences. The primary positive impact is enabling safer deployment of these models in real-world applications by providing better-calibrated confidence estimates, particularly crucial in high-stakes domains. However, we acknowledge that these techniques must be implemented thoughtfully to avoid potential misuse in creating deceptively confident models. We encourage practitioners to carefully consider these trade-offs when applying our methods.

## Acknowledgments

We would like to thank all the anonymous reviewers for their comments and suggestions. This work was conducted while Ruochen Jin was a visiting scholar at the University of Pennsylvania. This work was supported in part by NIH grants U01CA274576 and P30AG073105, ARPA-H Award D24AC00253, NSF grant DMS-2310679, a Meta Faculty Research Award, Wharton AI for Business, and a PSOM AI2D Seeding Project.

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

# A. Additional Discussion of Calibration

## A.1. Preference Optimization

**RLHF.** RLHF is an approach to align LLMs with human preferences, ensuring that the models are useful and safe for their human users (Ouyang et al., 2022). The framework for learning LLMs typically consists of the following three steps: (1) supervised fine-tuning (SFT), (2) Reward modeling, and (3) RLHF fine-tuning. The loss function of RLHF fine-tuning is

$$\max_{\phi} \mathbb{E}_{y \sim \pi_{\phi}(\cdot|x)} r(x, y) - \beta D_{\mathrm{KL}}(\pi_{\phi}(y|x) \| \pi_{\mathrm{ref}}(y|x)),$$

where $\beta > 0$ is a parameter controlling the deviation from the base reference policy $\pi_{\mathrm{ref}}$, namely the initial SFT model $\pi_{\mathrm{SFT}}$. $D_{\mathrm{KL}}$ is the Kullback–Leibler (KL) divergence between $\pi_{\phi}(y|x)$ and $\pi_{\mathrm{ref}}(y|x)$.

**DPO.** The original DPO method (Rafailov et al., 2023) is to directly optimize the policy without explicitly training the reward function in a supervised manner:

$$\mathbb{E}_{(x, y_w, y_l)} \log \sigma \left( \beta \log \frac{\pi_{\phi}(y_w|x)}{\pi_{\mathrm{ref}}(y_w|x)} - \beta \log \frac{\pi_{\phi}(y_l|x)}{\pi_{\mathrm{ref}}(y_l|x)} \right).$$

## A.2. Other Related Work

Several preference fine-tuning methods have been proposed for RLHF. Li et al. (2023b) showed that proximal policy optimization (PPO) (Schulman et al., 2017) does not fully exploit the potential of RLHF in aligning LLMs with human preferences. Tang et al. (2024) examined the performance gap between online and offline algorithms for alignment tasks, while Ye et al. (2024) introduced an online iterative RLHF algorithm that incorporates a general preference model. Li et al. (2025) studied the diversity of models trained during the SFT phase. Beyond fine-tuning, model editing (Jin et al., 2025) has emerged as another approach to adapt model behavior to various tasks.

Several notable variants of DPO have been developed (Liu et al., 2023; Azar et al., 2024; Chang et al., 2024; Gorbatovski et al., 2024; Rafailov et al., 2024a; Yang et al., 2024). However, recent studies (Li et al., 2023a; Xu et al., 2024a; Tajwar et al., 2024) suggest that DPO is less effective than reward-based RLHF methods for aligning LLMs. Both Li et al. (2023a) and Xu et al. (2024a) attributed this shortcoming to representation misspecification in DPO, which limits its ability to achieve robust alignment compared to reinforcement learning approaches such as PPO. Additionally, the on-policy nature of reward-based fine-tuning helps mitigate distribution shifts between the training data and online responses, thereby enhancing LLM performance (Tajwar et al., 2024).

Nash learning from human feedback (NLHF) has also been proposed as a framework to align LLMs with general preference models (Munos et al., 2023; Liu et al., 2025; Shi et al., 2025). Moreover, Wang et al. (2025) analyzed the convergence behavior of NLHF, showing that it approaches the Nash equilibrium in the last iterate.

Prior work (Guo et al., 2017; Müller et al., 2019) has shown that temperature scaling is generally more effective than label smoothing and other techniques for improving calibration. Zhao et al. (2021) proposed a contextual calibration procedure to improve few-shot performance of language models. Predictive accuracy and calibration trade-off have been studied in general machine learning classification problems (Kumar et al., 2018; Krishnan & Tickoo, 2020; Karandikar et al., 2021; Popordanoska et al., 2022). Additionally, several studies have provided theoretical analyses of the bias in ECE estimators based on uniform-mass binning (UMB) (Gupta et al., 2020; Gupta & Ramdas, 2021). Other related works include Gruber & Buettner (2022) and Sun et al. (2023), which explore the framework of proper calibration errors and minimum-risk recalibration of classifiers.

While temperature scaling (TS) is a strong baseline, our method's superiority stems from addressing fundamental limitations of post-hoc calibration in LLMs:

- **Direct optimization of calibration:** Our approach explicitly minimizes the discrepancy between accuracy and confidence—the definition of ECE—rather than relying on a single scaling parameter. This direct optimization improves calibration while maintaining or enhancing performance. This aligns with previous work on training classifiers with calibration objectives, such as the AvUC loss in Krishnan & Tickoo (2020) and the PAC-Bayes-based objective in Fujisawa & Futami (2024), which demonstrated the superiority of optimization-based strategies over post-hoc methods.

- **Improved generalization under distribution shift:** Our approach generalizes better to unseen data and distribution shifts compared to TS, which risks overfitting to specific validation sets. Table 2 clearly demonstrates this advantage in out-domain scenarios—for example, with Olmo2-7B, our CFT method reduces out-domain cw-ECE to 0.0637 (vs. TS's 0.1196) while improving accuracy to 0.7085 (vs. DPO's 0.6635). Distribution shifts are common in language tasks, and TS is insufficient to handle such variations.

## A.3. Multiclass Calibration

**Definition A.1** (Multiclass Calibration). A probabilistic classifier $\hat{\boldsymbol{p}} : \mathcal{X} \to \Delta_k$ is multiclass-calibrated, or simply calibrated, if for any prediction vector $\boldsymbol{q} = (q_1, \ldots, q_k) \in \Delta_k$, the proportions of classes among all possible instances $x$ getting the same prediction $\hat{\boldsymbol{p}}(x) = \boldsymbol{q}$ are equal to the prediction vector $\boldsymbol{q}$:

$$\mathbb{P}(y = j | \hat{\boldsymbol{p}}(x) = \boldsymbol{q}) = q_j, \quad j = 1, 2, \ldots k. \tag{7}$$

A widely used metric for evaluating confidence calibration is the Expected Calibration Error (ECE) Naeini et al. (2015), which is defined as the expected absolute difference between the model's confidence and its accuracy. Mlticlass-ECE (mc-ECE) is defined as:

$$\text{mc-ECE} = \mathbb{E}_{\hat{\boldsymbol{p}}(x)} \frac{1}{k} \sum_{j=1}^{k} \left| \mathbb{P}(y = j | \hat{\boldsymbol{p}}(x)) - \hat{p}_j(x) \right|.$$

**Relationships Between Calibration Types.** The following relationships hold:

- Multiclass calibration implies classwise calibration and confidence calibration.
- Classwise calibration implies confidence calibration.

## A.4. Rank Preserving Mapping

We denote the model confidence vector as $\text{Conf}_\pi(x) = [c_1, c_2, c_3, c_4]$, where $c_1 \geq c_2 \geq c_3 \geq c_4$ without loss of generality. Let the calibrated confidence be $\boldsymbol{p}(x) = [c_1', c_2', c_3', c_4']$. We set the top-1 calibrated confidence to match the bin accuracy:

$$c_1' = q_m.$$

To preserve the rank ordering of all four options, we aim to compress the remaining three confidence values $c_2, c_3, c_4$ into the interval $(0, q_m)$ while ensuring their sum equals $1 - q_m$. We apply a nonlinear transformation followed by a linear scaling:

$$c_i' = \alpha \tanh(\gamma c_i) + \beta, \quad \text{for } i = 2, 3, 4.$$

**Determining $\gamma$** To control the saturation level of the non-linear function $\tanh(\cdot)$, we select $\gamma$ such that the largest among $c_2, c_3, c_4$ maps close to 1:

$$\tanh\left(\gamma \max\{c_2, c_3, c_4\}\right) = 0.99.$$

To ensure numerical stability, we avoid $\tanh^{-1}(1)$ and use the approximation $\text{artanh}(0.99) \approx \ln(3)$. This gives:

$$\gamma \max\{c_2, c_3, c_4\} = \ln(3), \quad \Rightarrow \quad \gamma = \frac{\ln(3)}{\max\{c_2, c_3, c_4\}} \cdot \frac{1}{1 - q_m}.$$

**Solving for $\alpha$ and $\beta$** To satisfy the constraint that $\sum_{i=2}^{4} c_i' = 1 - q_m$, define:

$$T := \sum_{i=2}^{4} \tanh(\gamma c_i).$$

Then:

$$\sum_{i=2}^{4} c_i' = \sum_{i=2}^{4} \left(\alpha \tanh(\gamma c_i) + \beta\right) = \alpha T + 3\beta = 1 - q_m. \tag{8}$$

We can now solve for $\alpha$ and $\beta$ in two ways:

**(a) Simplified assumption:** $\beta = \alpha$

Substitute into Equation (8):

$$\alpha T + 3\alpha = 1 - q_m, \tag{9}$$
$$\Rightarrow \quad \alpha(T+3) = 1 - q_m, \tag{10}$$
$$\Rightarrow \quad \alpha = \frac{1-q_m}{T+3}, \quad \beta = \alpha. \tag{11}$$

**(b) General solution (no assumption):**

From Equation (8), isolate $\beta$:

$$\alpha T + 3\beta = 1 - q_m, \tag{12}$$
$$\Rightarrow \quad \beta = \frac{1 - q_m - \alpha T}{3}. \tag{13}$$

Thus, given any $\alpha$, $\beta$ can be directly computed.

**Ensuring Rank Preservation**   To preserve the ordering $c_i' < q_m$ for $i = 2, 3, 4$, we enforce:

$$c_i' = \alpha \tanh(\gamma c_i) + \beta < q_m.$$

Since $\tanh(\gamma c_i) < 1$, a sufficient condition is:

$$\alpha + \beta < q_m.$$

Substitute the simplified solution $\beta = \alpha$:

$$\alpha + \beta = 2\alpha < q_m.$$

Then:

$$2\alpha < q_m \quad \Rightarrow \quad \alpha < \frac{q_m}{2}, \tag{14}$$
$$\text{and since } \alpha = \frac{1-q_m}{T+3}, \quad \Rightarrow \quad \frac{1-q_m}{T+3} < \frac{q_m}{2}. \tag{15}$$

Multiply both sides of Equation (15) by $(T+3)$:

$$2(1 - q_m) < q_m(T+3), \tag{16}$$
$$2 < q_m(T+5), \tag{17}$$
$$\Rightarrow \quad q_m > \frac{2}{T+5}. \tag{18}$$

This inequality gives a general bound on $q_m$ for the mapping to preserve the ordering. Noting that $T = \sum_{i=2}^{4} \tanh(\gamma c_i) < 3$, we conservatively estimate $T \le 3$:

$$q_m > \frac{2}{T+5} \ge \frac{2}{8} = 0.25.$$

**Conclusion**   When $q_m > 0.25$, the parameters $\alpha$ and $\beta$ can be chosen such that the transformed confidences $c_i'$ satisfy $c_i' < q_m$ and preserve the original rank ordering. When $q_m \le 0.25$, the non-linear mapping using $\tanh(\gamma \cdot)$ compresses all $c_i'$ (for $i = 2, 3, 4$) near 1, effectively reflecting high uncertainty with low distinguishability across non-top predictions.

## A.5. Convergence of EM Algorithms

The convergence of EM algorithms is well studied; see, for example, Wu (1983). To ensure the convergence of the EM algorithm in our setting, three additional assumptions are required.

First, for the ECE loss,

$$\mathcal{L}_{\mathrm{ECE}} = \mathrm{D}(\boldsymbol{p}(x), \mathrm{conf}_\pi(x)),$$

we assume that $D$ is the cross-entropy loss. In this case,

$$\mathcal{L}_{\mathrm{ECE}} = -\mathbb{E}_{\boldsymbol{p}(x)}\left[\log \mathrm{conf}_\pi(x)\right].$$

Under this assumption, the loss function becomes a negative log-likelihood, which aligns with standard convergence theory for EM algorithms. However, in practice, alternative distance measures may also be used.

Second, the binary choice setting is also required. In this setting, the probability of the false answer is uniquely determined once the probability of the true answer is given. However, in the multiple-choice setting, the generative model assigns probability only to the correct answer, without imposing any constraints on the probabilities of the remaining choices. As a result, these probabilities do not influence the loss function, and the EM algorithm is not uniquely defined in this case. Therefore, the convergence guarantee only applies in the binary setting. In practice, we can use the mapping discussed above to assign probabilities to the remaining choices.

Finally, the optimization is performed over the choice probabilities rather than the neural network parameters, as the convergence of training deep neural networks remains an open problem.

# B. Proof of Technical Results

## B.1. Proof of Proposition 4.1

*Proof.* Based on the definition of the probabilistic generative model $\boldsymbol{p}$, for all $j$ and $x$, we have

$$\mathbb{P}(y = j | \boldsymbol{p}(x)) = p_j(x).$$

Then,

$$\mathrm{mc\text{-}ECE}(\boldsymbol{p}) = \mathbb{E}_x \frac{1}{k} \sum_{j=1}^{k} \left| \mathbb{P}(y = j | \boldsymbol{p}(x)) - p_j(x) \right| = 0.$$

$\square$

## B.2. Proof of Theorem 4.4

*Proof.* We first recap the definition of TCE:

$$\mathrm{TCE} = \mathbb{E}_x \frac{1}{k} \sum_{j=1}^{k} \left| p_j^*(x) - \hat{p}_j(x) \right|.$$

We partition all the samples into two subsets.

$$\mathcal{S}_{\mathrm{T}} = \{x | \arg\max \pi^*(x) = y\}.$$

$$\mathcal{S}_{\mathrm{F}} = \{x | \arg\max \pi^*(x) \neq y\}.$$

Denote $\mathrm{ACC}(\pi^*) = a^*$. We first assume that $a^* \leq a$. We then partition $\mathcal{S}_{\mathrm{F}}$ in to two parts: $\mathcal{S}_{\mathrm{F},1}$ and $\mathcal{S}_{\mathrm{F},2}$, with

$$\mathbb{P}\{x \in \mathcal{S}_{\mathrm{F},1}\} = a - a^*,$$

and

$$\mathbb{P}\{x \in \mathcal{S}_{\mathrm{F},2}\} = 1 - a.$$

Then, We define $\pi(x)$ as follows. When $x \in \mathcal{S}_{\text{T}}$ or $\mathcal{S}_{\text{F},2}$,

$$\pi(x) = \pi^*(x).$$

In this case, $\frac{1}{k} \sum_{j=1}^{k} \left| p_j^*(x) - \hat{p}_j(x) \right| = 0$.

When $x \in \mathcal{S}_{\text{F},1}$,

$$\pi_j(x) = \begin{cases} 1 & \text{if } j = y; \\ 0 & \text{if } j \neq y. \end{cases}$$

In this case, $\frac{1}{k} \sum_{j=1}^{k} \left| p_j^*(x) - \hat{p}_j(x) \right| = 1 - p_y^*(x) + \sum_{j \neq y} p_j^*(x) = 2 - 2p_y^*(x) \leq 2$.

If $a^* > a$. We then partition $\mathcal{S}_{\text{T}}$ in to two parts: $\mathcal{S}_{\text{T},1}$ and $\mathcal{S}_{\text{T},2}$, with

$$\mathbb{P}\{x \in \mathcal{S}_{\text{T},1}\} = a - a^*,$$

and

$$\mathbb{P}\{x \in \mathcal{S}_{\text{T},2}\} = 1 - a.$$

Then, We define $\pi(x)$ as follows. When $x \in \mathcal{S}_{\text{F}}$ or $\mathcal{S}_{\text{T},2}$,

$$\pi(x) = \pi^*(x).$$

In this case, $\frac{1}{k} \sum_{j=1}^{k} \left| p_j^*(x) - \hat{p}_j(x) \right| = 0$.

When $x \in \mathcal{S}_{\text{T},1}$,

$$\pi_j(x) = \begin{cases} 0 & \text{if } j = y; \\ 1 & \text{if } j \neq y. \end{cases}$$

In this case, $\frac{1}{k} \sum_{j=1}^{k} \left| p_j^*(x) - \hat{p}_j(x) \right| = 1 - p_y^*(x) + \sum_{j \neq y} p_j^*(x) = 2 - 2p_y^*(x) \leq 2$.

Therefore, for both of the two cases, we have

$$\text{TCE}(\pi) \leq 2|a - a^*| = 2|\text{ACC}(\pi^*) - \text{ACC}(\pi)|.$$

$\square$

## B.3. Proof of Theorem 4.5

*Proof.* Denote $\text{ACC}(\pi^*) = a^*$ and $\text{ACC}(\pi) = a$. By the definition of accuracy,

$$\mathbb{P}(x | \arg\max \pi^*(x) \neq \arg\max \pi(x)) \geq |a - a^*|.$$

For these $x$,

$$\frac{1}{k} \sum_{j=1}^{k} \left| p_j^*(x) - \hat{p}_j(x) \right| \geq \frac{1}{k} \max_{j=1,\dots,k} \left| p_j^*(x) - \hat{p}_j(x) \right| > 0.$$

Let

$$C = \frac{1}{k} \min_{\{x | \arg\max \pi^*(x) \neq \arg\max \pi(x)\}} \max_{j=1,\dots,k} \left| p_j^*(x) - \hat{p}_j(x) \right|.$$

We can see that $C > 0$ and

$$\text{TCE}(\pi) \geq C|a - a^*|,$$

for all $\pi$.

$\square$

## B.4. Proof of Theorem 4.6

*Proof.* We first recap the definition of cw-ECE:

$$\text{cw-ECE} = \mathbb{E}_{\hat{\boldsymbol{p}}(x)} \frac{1}{k} \sum_{j=1}^{k} \big| \mathbb{P}(y = j | \hat{p}_j(x)) - \hat{p}_j(x) \big|.$$

For any $q \in [0, 1]$:

$$\mathbb{P}(y = j | \hat{p}_j(x) = q) = \mathbb{E}_{\hat{p}_j(x) = q}[p_j^*(x)].$$

By a triangle inequality, we have

$$\big| \mathbb{P}(y = j | \hat{p}_j(x)) - \hat{p}_j(x) \big| \leq \mathbb{E}_{\hat{p}_j(x)} | p_j^*(x) - \hat{p}_j(x) |.$$

Therefore, we obtain that

$$\text{cw-ECE} \leq \text{TCE}.$$

$\square$

# C. Additional Experimental Details

## C.1. Comprehensive Descriptions for Models, Baselines, Datasets, and Metrics

**Models**   In our study, we employ four widely-used open-source large language models to investigate the calibration issue and validate the effectiveness of our proposed method. They include

- LLaMA-3.1-Tulu-8B-DPO[4]: LLaMA-3.1-Tulu-8B-DPO (Lambert et al., 2024) is a state-of-the-art instruction-following model developed by Allen Institute for AI. It is part of the Tulu 3 family, which is designed for diverse tasks beyond chat, such as MATH, GSM8K, and IFEval. The model is trained using supervised fine-tuning (SFT) and Direct Preference Optimization (DPO), achieving competitive performance on benchmarks like MMLU and TruthfulQA. It is fully open-source, with data, code, and training recipes available for reproducibility.
- Vicuna-7B-v1.5[5]: Vicuna-7B-v1.5 (Chiang et al., 2023) is a chat assistant developed by LMSYS, fine-tuned from Llama 2 using approximately 125,000 user-shared conversations collected from ShareGPT.com. This auto-regressive language model employs the transformer architecture and is designed for research purposes in natural language processing, machine learning, and artificial intelligence. The model has been evaluated using standard benchmarks, human preferences, and LLM-as-a-judge methodologies.
- Olmo 2-7B-DPO[6]: Olmo 2-7B-DP (OLMo et al., 2024) is a fully open-source language model from Allen Institute for AI, designed for research and educational use. It is trained on the Dolma dataset and fine-tuned using DPO for improved performance on tasks like text generation and instruction following. The model is part of the OLMo series, which emphasizes transparency by releasing weights, data, and training details. It achieves strong results on benchmarks such as GSM8K and MATH.
- Mistral 7B-DPO[7]: Mistral 7B-DPO (Jiang et al., 2023a) is a high-performance language model fine-tuned using Direct Preference Optimization. It is designed for tasks like text generation, instruction following, and reasoning. The model is part of the Mistral family, known for its efficiency and strong performance on benchmarks such as HumanEval and GSM8K. It is widely used in research and applications requiring robust natural language understanding.

Given the limitation of our computational resources (only four A100 (40G) GPUs), we utilize the Quantized Low Rank (QLoRA) technique (Dettmers et al., 2024) to optimize our workflow. It is worth noting that all selected models are aligned with human values according to either Reinforcement Learning with Human Feedback (RLHF) (Christiano et al., 2017) or Direct Preference Optimization (DPO) (Rafailov et al., 2024b), providing a robust framework for alignment and adaptability in our experimental evaluations.

---

[4] https://huggingface.co/allenai/Llama-3.1-Tulu-3-8B
[5] https://huggingface.co/lmsys/vicuna-7b-v1.5
[6] https://huggingface.co/allenai/OLMo-2-1124-7B-DPO
[7] https://huggingface.co/princeton-nlp/Mistral-7B-Base-SFT-DPO

**Baselines**   We validate the effectiveness of the proposed method by comparing performance among four kinds of models.

- The model after alignment with human preference: This kind of model is typically aligned with human preference via RLHF or DPO.
- The model calibrated by Temperature Scaling (TS) (Guo et al., 2017): TS adjusts the confidence scores of a pre-trained model by introducing a single parameter called the "temperature" (denoted as $T$). This parameter scales the logits (outputs before the softmax function) to produce softer probability distributions. The temperature $T$ is searched on a validation set to directly minimize the ECE and thus the model calibated by TS is usually well-calibrated.
- The model fine-tuned by CFT (ours): This kind of model is calibrated by our method CFT where we only use the completion or response to calculate the loss. In this way the model learns patterns rather than knowledge. This approach does not improve generalization (hence, accuracy remains largely unchanged), but it mitigates the model's overconfidence by focusing on patterns that influence confidence. As a result, the ECE is significantly reduced.
- The model fine-tuned by Regularized CFT (ours): This kind of model is calibrated by our method Regularized CFT (RCFT (Ours)) where we use both the prompt (questions and options) and the completion (response) to calculate the loss. In this way, the model learns knowledge from the prompt, improving its generalization ability. This leads to higher accuracy on the test set but also keeps the model in an overconfident state. To address this, additional calibration loss is required to adjust the model's confidence.

**Dataset**   To evaluate the efficacy of our proposed calibration method, we employ five datasets to conduct comprehensive experiments:

- MMLU (Massive Multitask Language Understanding)[8]: MMLU is a benchmark dataset designed to evaluate the knowledge and reasoning capabilities of language models across 57 subjects, ranging from STEM to humanities and social sciences. It includes questions at various difficulty levels, from elementary to advanced professional, and is particularly useful for assessing zero-shot and few-shot learning performance. The dataset is structured to test both world knowledge and problem-solving abilities, making it a comprehensive tool for identifying model blind spots.
- MedMCQA[9]: MedMCQA is designed for medical multiple-choice question (MCQs), which includes a comprehensive collection of advanced-level questions covering various medical fields such as Anesthesia, Anatomy, Biochemistry, and more. The dataset comprises approximately 194k MCQs, sourced from AIIMS and NEET PG entrance exams. We randomly select 5000 samples, with 3500 used as the training set for fine-tuning and 1500 as the testing set for inference
- OpenBookQA[10]: OpenBookQA is a dataset modeled after open-book exams, designed to assess the understanding and application of core scientific facts. It consists of 5,957 elementary-level science questions, each linked to a small "book" of 1,326 core science facts. The dataset requires models to apply broad common knowledge beyond the provided facts, making it challenging for retrieval-based and word co-occurrence algorithms. It includes 4,957 training, 500 development, and 500 test questions.
- ARC-Challenge[11]: The ARC-Challenge dataset is a collection of 2,590 multiple-choice science questions designed to test advanced knowledge and reasoning skills. These questions are derived from science exams for grades 3 through 9 and are specifically curated to be challenging for both humans and AI systems, with each question having four answer choices. The questions require a deep understanding of scientific concepts, logical reasoning, and the ability to infer relationships between ideas. The ARC-Challenge is widely used as a benchmark to evaluate the performance of AI models in complex question-answering tasks, pushing the boundaries of natural language understanding and reasoning capabilities.
- AlpacaEval[12]: Evaluation of instruction-following models (e.g., ChatGPT) typically requires human interaction, which is time-consuming, expensive, and difficult to replicate. `AlpacaEval` is an LLM-based automatic evaluation framework that is fast, cost-effective, reproducible, and validated against 20K human annotations. It is particularly useful for model development. Although it improves upon prior automatic evaluation pipelines, `AlpacaEval` still exhibits fundamental limitations, such as a preference for longer outputs. The framework provides the following components:
    - Leaderboard: A leaderboard reporting the performance of common models on the `AlpacaEval` evaluation set.

---

[8]https://huggingface.co/datasets/cais/mmlu
[9]https://github.com/medmcqa/medmcqa
[10]https://huggingface.co/datasets/allenai/openbookqa
[11]https://huggingface.co/datasets/allenai/ai2_arc
[12]https://github.com/tatsu-lab/alpaca_eval

Caution: Automatic evaluators (e.g., GPT-4) may be biased toward models that produce longer outputs or were fine-tuned on the same base model as the evaluator (e.g., GPT-4).

- Automatic Evaluator: An evaluator with high agreement with human judgments (validated on 20K annotations). It measures performance by computing the fraction of times a strong LLM (e.g., GPT-4) prefers the outputs from the evaluated model over those from a reference model. The evaluator supports output caching and randomization by default.
- Toolkit for Building Evaluators: A simple interface for constructing advanced automatic evaluators with features such as caching, batching, multi-annotator support, and statistical analysis (e.g., quality, price, speed, statistical power, bias, and variance).
- Human Evaluation Data: A dataset of 20K human preferences comparing outputs from a given model and a reference model on the `AlpacaFarm` evaluation set. This includes 2.5K cross-annotations, where four human annotators rated the same 650 examples.
- AlpacaEval Dataset: A simplified version of the `AlpacaFarm` evaluation set, where "instructions" and "inputs" are merged into a single field and reference outputs are extended in length. See the documentation for further details.

- Arena-Hard[13]: `Arena-Hard-Auto` is an automatic evaluation tool for instruction-tuned LLMs. Among popular open-ended LLM benchmarks, `Arena-Hard-Auto` demonstrates the highest correlation and separability with `LMArena` (Chatbot Arena); see the associated paper for details. If you are interested in estimating how well your model might perform on `LMArena` prior to deployment, we recommend using the latest evaluation set, `Arena-Hard-v2.0-Preview`.
- UltraFeedback-Binarized-Preferences[14]: This dataset, hosted on Hugging Face by Argilla, is a resource designed to support research in preference modeling and reinforcement learning from human feedback (RLHF). This dataset provides a collection of binary-labeled preferences derived from UltraFeedback data, where each entry represents a human judgment indicating preference between two competing outputs. It is tailored for fine-tuning models to align with human-like decision-making and to enable evaluation of performance in feedback-based ranking tasks. By offering structured preference annotations, it empowers researchers to develop and evaluate models that incorporate nuanced human feedback, facilitating advancements in AI alignment and personalization.

These datasets are instrumental in assessing the calibration performance as well as the model's original language ability.

**Metric** Here we introduce three metrics used in our paper: The discrete conf-ECE and cw-ECE, and Win Rate. The discrete conf-ECE and cw-ECE are defined as follows.

**Confidence Expected Calibration Error (conf-ECE)** is defined as

$$\text{conf-ECE} = \sum_{m=1}^{M} \frac{|B_m|}{N} |\mathbb{P}(Y = I|\mathbf{x} \in B_m) - \mathbb{E}[\pi_\theta(\mathbf{y} = I|\mathbf{x})|\mathbf{x} \in B_m]|, \tag{19}$$

where $B_m$ is the $j$-th bin in terms of max confidence; $|B_m|$ denotes the cardinality of the bin; $\mathbb{P}(Y = I|\mathbf{x} \in B_m)$ and $\mathbb{E}[\pi_\theta(\mathbf{y} = I|\mathbf{x})|\mathbf{x} \in B_m]$ denote the average prediction of winning class probability and the actual probability that the winning class is the ground truth class.

**Class-Wise Expected Calibration Error (cw-ECE)** is defined as

$$\text{cw-ECE} = \frac{1}{K} \sum_{j=1}^{k} \sum_{m=1}^{M} \frac{|B_{m,j}|}{N} |\mathbb{P}(Y = i|\mathbf{x} \in B_{m,j}) - \mathbb{E}[\pi_\theta(\mathbf{y} = i|\mathbf{x})|\mathbf{x} \in B_{m,j}]|, \tag{20}$$

where $B_{m,j}$ is the $m$-th bin of the $j$-th class; $|B_{m,j}|$ denotes the cardinality of the bin; $\mathbb{P}(Y = j|\mathbf{x} \in B_{m,j})$ and $\mathbb{E}[\pi_\theta(\mathbf{y} = j|\mathbf{x})|\mathbf{x} \in B_{m,j}])$ denote the average prediction of bin $j$ probability and the actual proportion of bin $j$ in the bin $B_{m,j}$.

We then introduce the detail of the **Win Rate** metric. **Win Rate** metric evaluates the model's "language ability" in two aspects.

---

[13] https://github.com/lmarena/arena-hard-auto
[14] https://huggingface.co/datasets/argilla/ultrafeedback-binarized-preferences

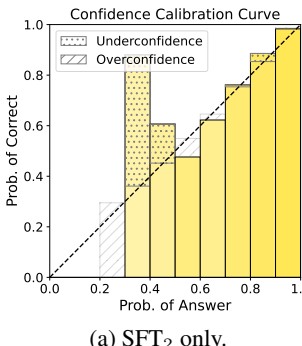
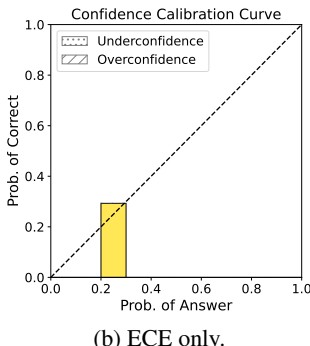

(a) SFT$_2$ only.                    (b) ECE only.

*Figure 4.* (a) Using only $\mathcal{L}_{\text{SFT}_2}$ achieves high accuracy ($\sim$90%) but shows poor calibration. (b) Using only $\mathcal{L}_{\text{ECE}}$ drives the model toward random guessing, achieving near-zero ECE but with only 25% accuracy.

The first one is the ability to distinguish between preferred and non-preferred responses within a given dataset. To this end, we utilize a widely-used dataset UltraFeedback [15] which provides pairs of responses: a *chosen_response* (preferred) and a *reject_response* (non-preferred) for each instruction. We then calculate the *sequence probabilities* of these provided responses under the target model's distribution. The **sequence probability** of a response $R = (w_1, w_2, \ldots, w_T)$, where $w_i$ denotes the $i$-th token, is computed as:

$$P(R) = P(w_1, w_2, \ldots, w_T) = \prod_{i=1}^{T} P(w_i \mid w_1, w_2, \ldots, w_{i-1}). \tag{21}$$

Here, $P(w_i \mid w_1, w_2, \ldots, w_{i-1})$ represents the probability of token $w_i$ given its preceding context, as determined by the model. For each instruction in the dataset, we calculate the sequence probabilities $P_{\text{chosen}}$ and $P_{\text{reject}}$ for the *chosen_response* and *reject_response*, respectively. A "win" is recorded if $P_{\text{chosen}} > P_{\text{reject}}$; otherwise, it is a "loss."

The second aspect is the quality of the response. We use the questions from two popular datasets AlpacaEval[16] and Arena-Hard[17] and let our model and a baseline model (usually GPT-4/GPT-4o or the DPO/RLHF model that is well aligned with human preference) make two separate responses. We then let a third party large language model (such as GPT-4 or GPT-4o) judge which one is better. A "win" of our model is recorded if the judge thinks its response is better than the baseline; otherwise, it is a "loss."

The **Win Rate** is then defined as the proportion of wins over the total number of comparisons:

$$\text{Win Rate} = \frac{\text{Number of Wins}}{\text{Total Number of Comparisons}}.$$

We anticipate that the model adjusted using our approach will achieve a competitive Win Rate in comparison to the model prior to calibration. This will demonstrate that the good alignment performance brought by DPO/RLHF will not be diminished when we lift the calibration performance.

### C.2. Additional Ablation Study

Figure 4 shows the ablation study in RCFT. In RCFT's objective, we have two components. One is the SFT loss $\mathcal{L}_{\text{SFT}_2}$ and the other is the calibration loss $\mathcal{L}_{\text{ECE}}$. We want to investigate the effect of each of them solely. Figure 4(a) shows the calibration plot of using $\mathcal{L}_{\text{SFT}_2}$ to optimize only. We can see that this loss function can significantly improve the accuracy but the calibration performance is destroyed. On the other hand, Figure 4(b) exhibits a very good calibration performance (the column is right on the perfect calibration diagonal line). However, its accuracy is very low (around 0.25), indicating the model is randomly guessing. RCFT combines these two loss functions together, resulting in a good trade-off between accuracy and calibration.

---

[15]https://huggingface.co/datasets/argilla/ultrafeedback-binarized-preferences
[16]https://github.com/tatsu-lab/alpaca_eval
[17]https://github.com/lmarena/arena-hard-auto

Table 4 presents the effect of varying the calibration regularization weight $\lambda$ in our fine-tuning objective. When $\lambda = 0$, the model corresponds to standard SFT, achieving the highest accuracy but relatively poor calibration. As $\lambda$ increases, both ECE and class-weighted ECE generally improve, indicating better alignment between model confidence and correctness. However, this comes with a noticeable drop in accuracy, particularly for higher values of $\lambda$, such as 1.8 or in the "ECE only" setting. The results highlight a trade-off between predictive accuracy and calibration, and suggest that moderate values of $\lambda$ (e.g., $\lambda = 1$) can strike a balance between the two.

*Table 4.* Ablation on calibration weight $\lambda$ in the loss function. We report the accuracy and calibration metrics (ECE and cw-ECE) for models fine-tuned with varying $\lambda$ in the objective. A higher $\lambda$ places more emphasis on calibration.

| Lambda | 0 | | | 0.4 | | | 1 | | | 1.8 | | | ECE only | | |
|---|---|---|---|---|---|---|---|---|---|---|---|---|---|---|---|
| Metric | ECE | cw-ECE | Acc | ECE | cw-ECE | Acc | ECE | cw-ECE | Acc | ECE | cw-ECE | Acc | ECE | cw-ECE | Acc |
| Llama-3.1-8B | 0.0883 | 0.0808 | 0.8964 | 0.1535 | 0.1014 | 0.8409 | 0.0897 | 0.0771 | 0.8341 | 0.0178 | 0.0106 | 0.4366 | 0.0002 | 0.0081 | 0.2475 |
| Vicuna-7B | 0.1219 | 0.0774 | 0.8322 | 0.1620 | 0.0991 | 0.7315 | 0.0474 | 0.0459 | 0.6015 | 0.1052 | 0.0799 | 0.3877 | 0.0130 | 0.0270 | 0.2290 |
| Olmo2-7B | 0.1003 | 0.0992 | 0.8846 | 0.1771 | 0.1008 | 0.8427 | 0.0989 | 0.0806 | 0.8510 | 0.0038 | 0.0113 | 0.4901 | 0.0030 | 0.0043 | 0.2765 |
| Mistral-7B | 0.0976 | 0.0785 | 0.9091 | 0.1316 | 0.0733 | 0.8085 | 0.0979 | 0.0877 | 0.8297 | 0.0366 | 0.0617 | 0.4217 | 0.0021 | 0.0108 | 0.2670 |

### C.3. Additional Experimental Results

Figure 5 and Figure 7 show the complete calibration plots on Llama-3.1-8B-Tulu. We can see that CFT and RCFT exhibit calibration improvement by providing good alignment with the *perfect calibration* diagonal line. In Figure 7, the columns are not be able to show a good alignment with the diagonal line, showing various behaviors in different options. After merging them into one figure (shown in Figure 5), the biases of the different options cancel each other out, resulting in a good classwise calibration. Figure 6 and Figure 8 are the calibration plots on Olmo-7B, which show similar phenomenon as Figure 5 and Figure 7, validating the effectiveness of the proposed methods.

### C.4. Discussion of Bin Size

Recent studies have shown that the estimation of ECE suffers from significant estimation bias (e.g., Futami & Fujisawa (2024) for binary classification; Fujisawa & Futami (2024) for multiclass classification). According to these works, binning-based ECE exhibits a slow convergence rate of $\mathcal{O}(n^{-1/3})$ and incurs substantial bias. According to the referenced work, the optimal bin size for the multiclass setting scales as $\mathcal{O}(n^{-1/3})$. However, to apply this practically, one needs the exact constant rather than just the asymptotic rate. Upon further examination, we found that the optimal bin size contains a Lipschitz constant of the model, in a rate of

$$\mathcal{O}(L^{2/3}n^{-1/3}).$$

In practice, the Lipschitz constant of deep neural networks is known to be very large; see, for example, our previous work (Xiao et al., 2023; 2024b;c). Estimating the Lipschitz constant of Transformer architectures in particular remains challenging.

In our experiments, the sample size is 3,000, which implies an optimal bin size on the order of $\mathcal{O}(14.4)$. As suggested by the anonymous reviewer, the value $n^{-1/3}$ plays an important role in estimating the ECE, and the actual ECE value can vary significantly depending on whether the number of bins is 10 or 14.

On this point, we hold a different view: we believe that the Lipschitz constant dominates the optimal convergence rate, and that the theoretical order is primarily of academic interest rather than of practical significance.

Nonetheless, we can still make use of the rate as in the theoretical papers. We adopt a bin size of 14 and evaluate all four methods across different architectures, ECE variants, and both in-domain and out-of-domain settings. The results are presented in Table 5.

We observe that CFT and RCFT consistently improve the ECE of DPO models. The comparison between our approach and TS remains consistent with our original findings: in 5 out of 8 conf-ECE comparisons, CFT outperforms TS.

## Use of Generative AI

The authors used generative LLMs only for proofreading, checking grammar, and correcting typos to improve the readability of the paper.

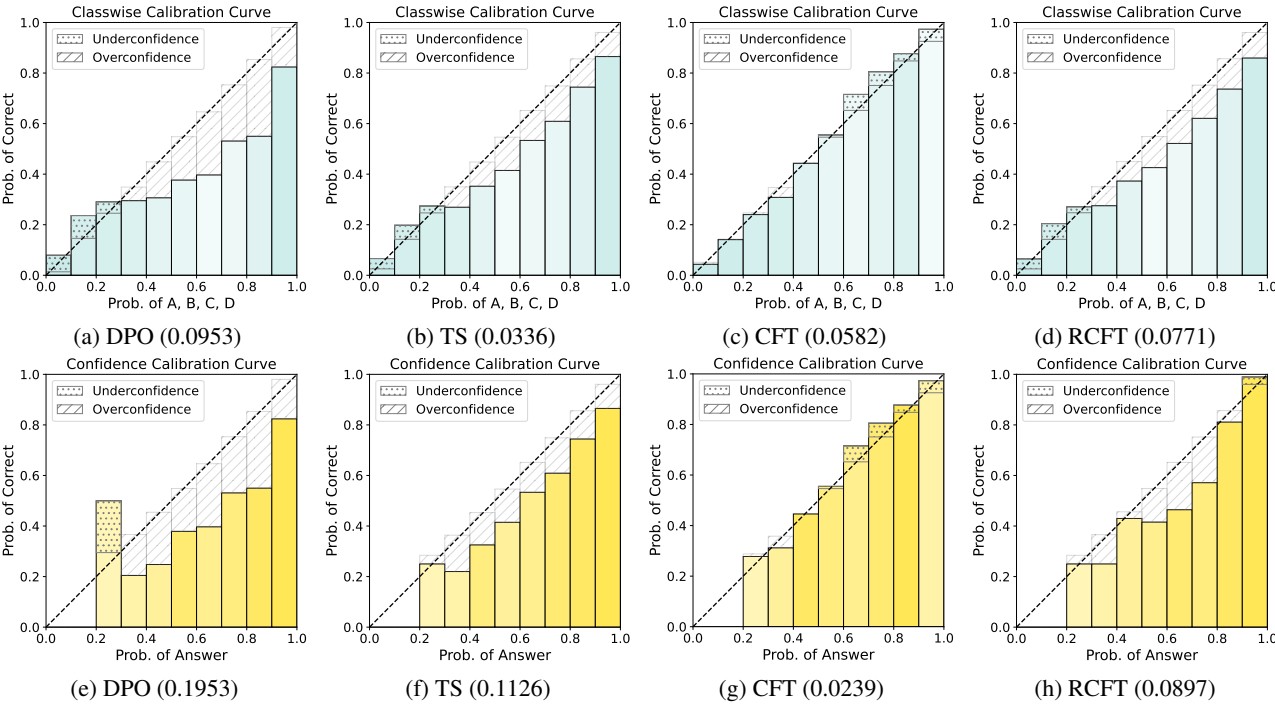

*Figure 5.* Calibration Plots of (a, e) DPO, (b, f) Temperature Scaling (TS), (c, g) our CFT, (d, h) our RCFT on Llama-3.1-8B-Tulu. (a-d) are the classwise calibration curve and (e-h) are the confidence calibration curve. Each panel plots the model's predicted probabilities (i.e., confidence) on the $x$-axis against the observed accuracy (fraction correct) on the $y$-axis, binned into ten groups. The diagonal line in each panel represents *perfect calibration*. The depth of the color indicates the sample density in that column. DPO has the worst calibration performance. Other three methods improve the calibration performance where our CFT has the lowest con-ECE (shown in the parenthesis). The figures of conf-ECE (e-h) omit the first two bins because the model selects an answer with the largest predicted probability which is always larger than 0.25 in the four options prediction task (so no samples exist below that threshold).

*Table 5.* Performance comparison among DPO/RLHF, Temperature Scaling, CFT, and RCFT across four models (**Llama3.1-8B-Tulu**, **Vicuna-7B**, **Olmo2-7B**, and **Mistral-7B**) in in-domain and out-domain scenarios. Best results in each metric block are bolded. "↓" means the smaller the better.

| Model | Method | conf-ECE ↓ | | cw-ECE ↓ | |
|---|---|---|---|---|---|
| | | In-Domain | Out-Domain | In-Domain | Out-Domain |
| Llama3.1-8B-Tulu | DPO | 0.1861 | 0.1188 | 0.0988 | 0.0657 |
| | Temp Scale. | 0.1158 | 0.0559 | **0.0349** | **0.0256** |
| | CFT(Ours) | **0.0441** | **0.0520** | 0.0418 | 0.0344 |
| | RCFT(Ours) | 0.1011 | 0.0801 | 0.0783 | 0.0525 |
| Vicuna-7B | RLHF | 0.1418 | 0.0888 | 0.0664 | 0.0993 |
| | Temp Scale. | 0.0377 | **0.0297** | **0.0220** | 0.0523 |
| | CFT(Ours) | **0.0216** | 0.0308 | 0.0295 | **0.0516** |
| | RCFT(Ours) | 0.0508 | 0.0677 | 0.0397 | 0.0552 |
| Olmo2-7B | DPO | 0.1370 | 0.0914 | 0.0773 | 0.0630 |
| | Temp Scale. | **0.0490** | **0.0272** | **0.0329** | **0.0252** |
| | CFT(Ours) | 0.0587 | 0.0573 | 0.0376 | 0.0356 |
| | RCFT(Ours) | 0.0730 | 0.0663 | 0.0365 | 0.0512 |
| Mistral-7B | DPO | 0.1979 | 0.1346 | 0.1010 | 0.1187 |
| | Temp Scale. | 0.0771 | 0.1093 | 0.0380 | 0.0582 |
| | CFT(Ours) | **0.0602** | **0.0511** | **0.0207** | 0.0601 |
| | RCFT(Ours) | 0.0817 | 0.0658 | 0.0457 | **0.0506** |

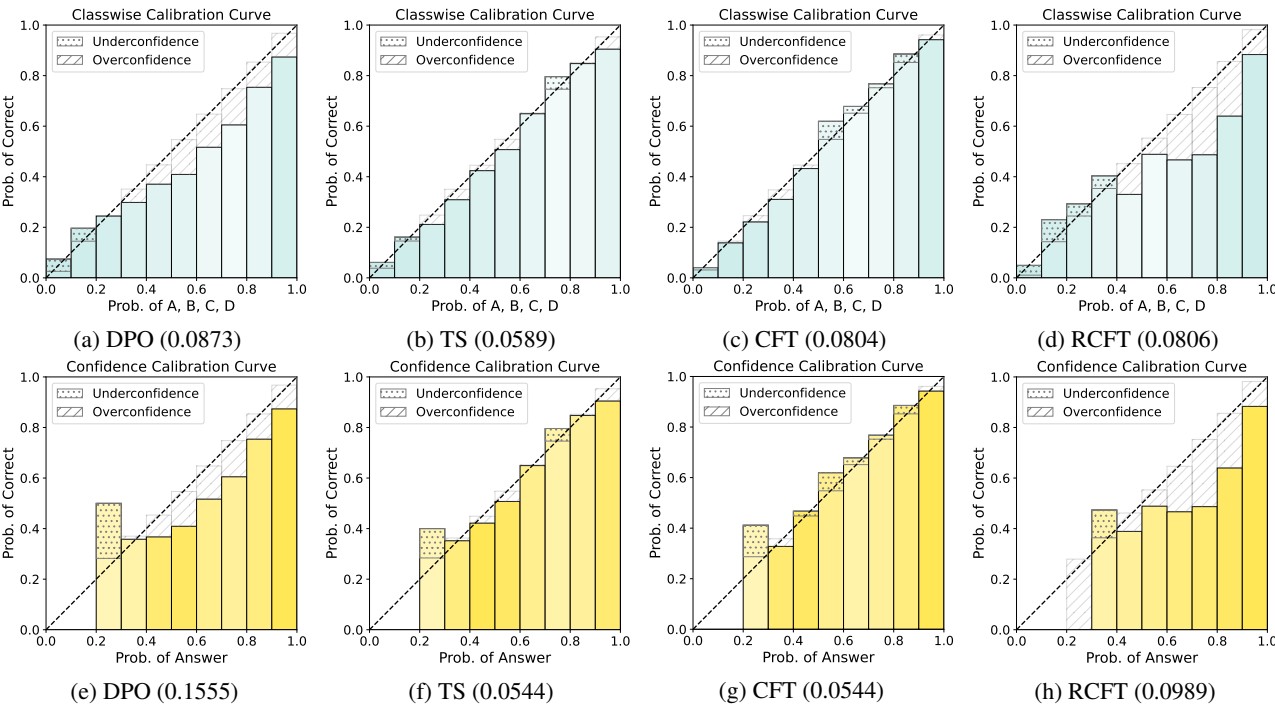

*Figure 6.* Calibration Plots of (a, e) DPO, (b, f) Temperature Scaling, (c, g) our CFT, and (d, h) our RCFT on Olmo2-7B. (a-d) are the classwise calibration curve and (e-h) are the confidence calibration curve. Each panel plots the model's predicted probabilities (i.e., confidence) on the $x$-axis against the observed accuracy (fraction correct) on the $y$-axis, binned into ten groups. The diagonal line in each panel represents *perfect calibration*. The depth of the color indicates the sample density in that column. DPO has the worst calibration performance. Other three methods improve the calibration performance where our CFT has the lowest con-ECE (shown in the parenthesis). The figures of conf-ECE (e-h) omit the first two bins because the model selects an answer with the largest predicted probability which is always larger than 0.25 in the four options prediction task (so no samples exist below that threshold).

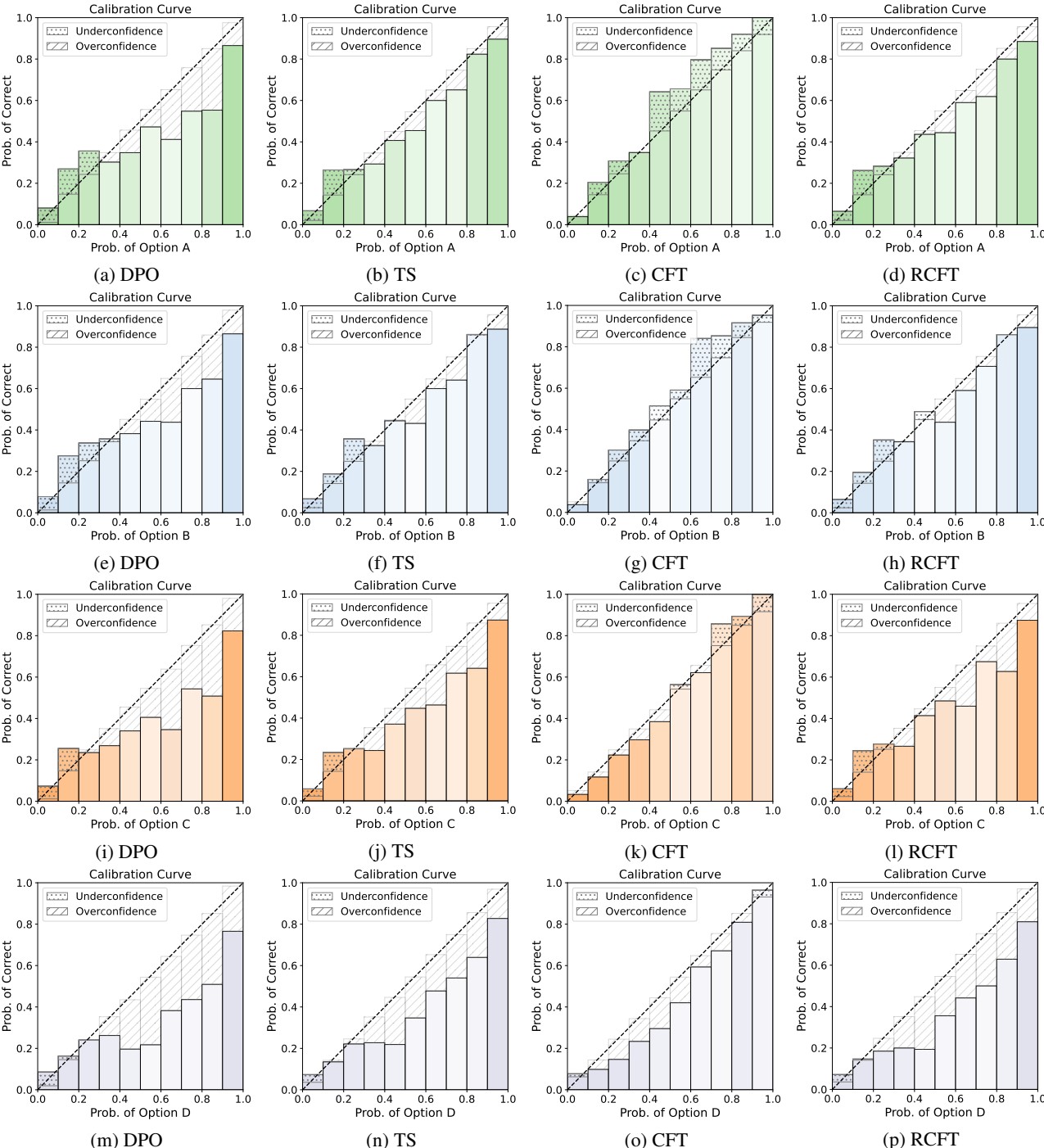

*Figure 7.* Calibration Plots of (a, e, i, m) DPO, (b, f, j, n) Temperature Scaling (TS), (c, g, k, o) our CFT, (d, h, l, p) our RCFT on Llama-3.1-8B-Tulu. Each panel plots the model's predicted probabilities (i.e., confidence) on the $x$-axis against the observed accuracy (fraction correct) on the $y$-axis, binned into ten groups. The diagonal line in each panel represents *perfect calibration*. The depth of the color indicates the sample density in that column.

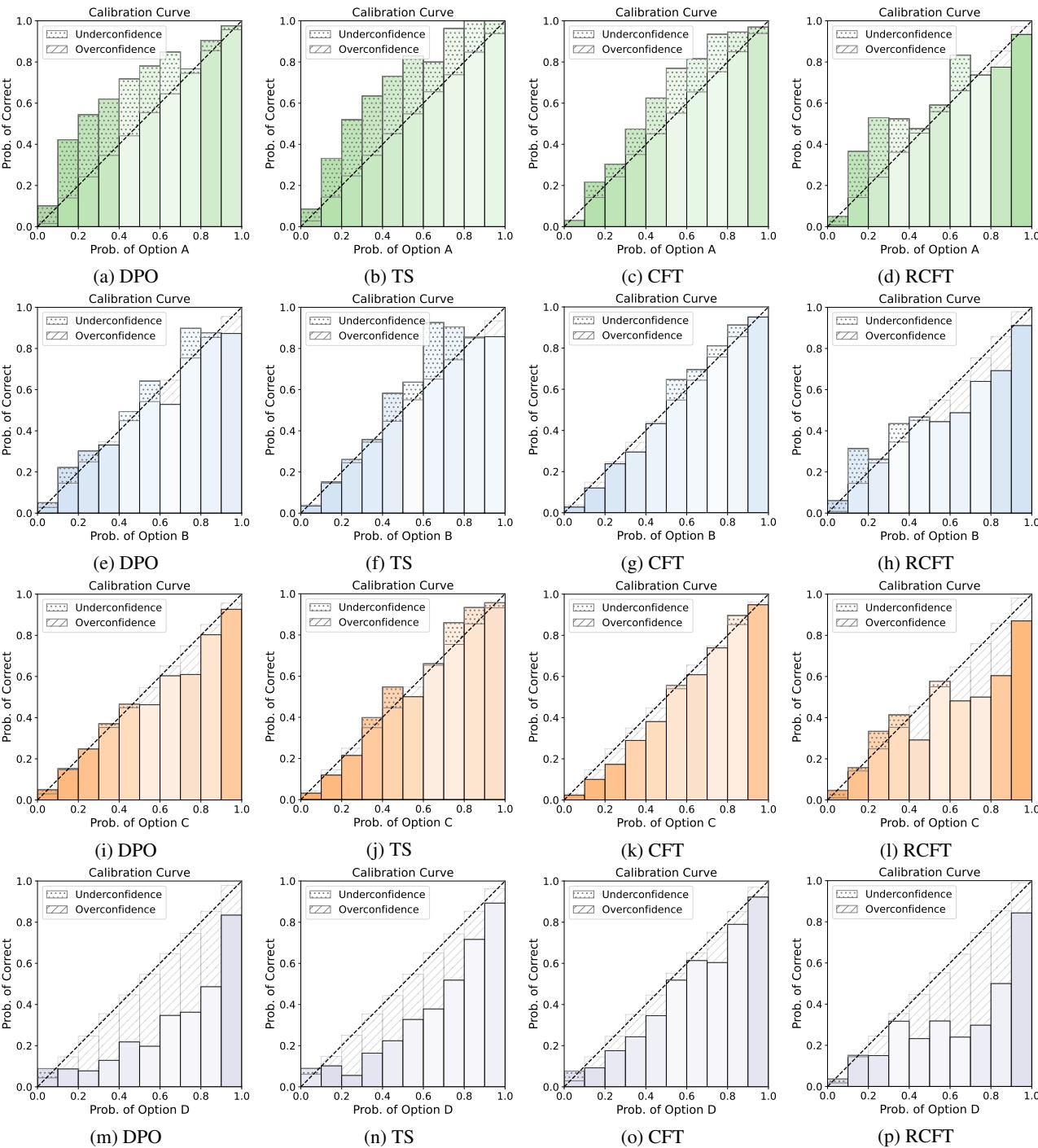

*Figure 8.* Calibration Plots of (a, e, i, m) DPO, (b, f, j, n) Temperature Scaling, (c, g, k, o) our CFT, and (d, h, l, p) our RCFT on Olmo2-7B. Each panel plots the model's predicted probabilities (i.e., confidence) on the $x$-axis against the observed accuracy (fraction correct) on the $y$-axis, binned into ten groups. The diagonal line in each panel represents *perfect calibration*. The depth of the color indicates the sample density in that column.

