# OpenReview forum: "Restoring Calibration for Aligned Large Language Models: A Calibration-Aware Fine-Tuning Approach"
_ICML.cc/2025/Conference — ICML 2025 poster_

### Official Review · Reviewer_hpMa · 2025-03-10

**Overall Recommendation:** 3

**Summary:**

The paper points out the problem of overconfidence of preference aligned Large Language Models (LLMs), and propose two fine-tuning approaches to address the problem: calibration-aware fine-tuning (CFT) and regularized CFT (RCFT).

**Claims And Evidence:**

The biggest problem is that though the authors state that preference alignment (like RLHF and DPO) causes poor calibration in LLMs, they do not propose any modifications to preference alignment, but instead propose fine-tuning approaches. As we know that Supervised Fine-Tuning (SFT) and RLHF are two different post-training stages, it is very essential for authors to justify why fine-tuning (rather than improving over the current RLHF framework) is a better solution, and should people use the proposed CFT before or after RLHF, or completely replace RLHF with CFT.

Moreover, it is an unfair comparison between an extra domain-knowledge fine-tuned LLM and an LLM not trained with domain knowledge. Intuitively the trained LLM will have more information on the in-domain tasks.

**Essential References Not Discussed:**

Improving over SFT:
[1] Enhancing confidence expression in large language models through learning from past experience. https://arxiv.org/abs/2404.10315.

Improving over RLHF:
[2] When to Trust LLMs: Aligning Confidence with Response Quality. ACL 2024. https://arxiv.org/abs/2404.17287
[3] Taming Overconfidence in LLMs: Reward Calibration in RLHF. ICLR 2025. https://arxiv.org/abs/2410.09724

**Experimental Designs Or Analyses:**

The experiments are rather complete on four tasks and the authors use a wide range of metrics: ECE, classwise-ECE, accuracy and win-rate.

However, it is unclear which stage the authors apply CFT. If they apply CFT after RLHF, then that means post-training of a pre-trained model will now include three steps: traditional SFT, RLHF, and CFT. This will create a very long training pipeline which is not practical in real practice, and authors need to discuss this limitation.

Since the authors propose to calibrate RLHF-trained models, it is also important to see if the calibration approach will hinder the original preference alignment ability, such as testing their models on benchmarks like Arena-Hard. Otherwise they are simply fine-tuning a model for a specific usage instead of generally improving the RLHF-trained models.

In fact, there are already previous works that improve LLM calibration through RLHF and maintain preference alignment ability [1][2], and they did a lot of study on whether improving the RLHF framework will hinder other generic capabilities.

[1] When to Trust LLMs: Aligning Confidence with Response Quality. ACL 2024. https://arxiv.org/abs/2404.17287
[2] Taming Overconfidence in LLMs: Reward Calibration in RLHF. ICLR 2025. https://arxiv.org/abs/2410.09724

**Methods And Evaluation Criteria:**

Theoretical explanation:
The authors propose calibratable and non-calibratable regime. It is better if they can explain at first that the difference of these two regime is that whether it is possible to achieve perfect calibration without sacrificing accuracy.

Proposed SFT method:
It is unclear why the proposed domain-specific fine-tuning in sec. 5.1 can help improve LLM calibration.
In both Sec 5.1 and 5.2, the authors no longer mention calibratable or non-calibratable regime at all. It is hard to tell when people should use methods in Sec 5.1, and when to use methods in Sec 5.2.

Evaluation criteria:
The authors use four RLHF-trained models and evaluate on a wide range of datasets.

**Other Comments Or Suggestions:**

N/A

**Other Strengths And Weaknesses:**

The paper writing is problematic:

1. The definition of multiple-choice question in Sec. 3 and Sec 4 is too narrow: only four-answer setting (A,B,C,D) is allowed. The authors can relax to a few number of choices, not just four.

2. Many grammar errors in paper writing, just to list a few:
"Another researchers teach the LLMs" -> "Other researchers teach..."
"The DPO method is to directly optimize of the policy without explicitly training the reward function" -> "The DPO method is to directly optimize the policy ..."

3. Figure 2. The description of y-axis is unclear, and the authors need to mention which "value" it is.

**Questions For Authors:**

1. What is the overall post-training pipeline when CFT is applied?
2. When to use CFT and when to use RCFT?
3. Will CFT training reduce model's preference alignment ability?

**Relation To Broader Scientific Literature:**

The authors echo with previous research that preference aligned LLMs are more overconfident. However, they are proposing an SFT-based approach to improve model calibration. There are of course methods directly improving over traditional SFT [1][2], but the authors in this paper are proposing a CFT method after the RLHF stage, they will need to justify the long post-training pipeline of: SFT -> RLHF -> CFT.

[1] Teaching Models to Express Their Uncertainty in Words. https://arxiv.org/abs/2205.14334.
[2] Enhancing confidence expression in large language models through learning from past experience. https://arxiv.org/abs/2404.10315.

**Theoretical Claims:**

The theoretical proofs looks good, though I think the authors should not restrict to only four possible answers (only four-answer setting (A,B,C,D) is allowed), as that is not practical in real applications.

---

> ### Author Rebuttal · Authors · 2025-04-01
>
> We thank reviewer hpMa for the comments and suggestions. Below we provide our answer to your questions.
>
> >Q1. The biggest problem is ... not propose any modifications to preference alignment.
>
> A. Thank you for the comment. We would like to clarify that this is better to be considered as a realistic practical scenario instead of a problem.
>
> To begin with, consider a practical deployment scenario, which is the primary focus of our work. A practitioner aiming to develop their own model typically starts by downloading an open-source LLM and adapting it to their specific needs. Importantly, for many such models, the intermediate checkpoints (e.g., SFT stages) are not publicly available. As a result, practitioners are often unable to modify or re-run the full preference alignment process. In such cases, post-alignment fine-tuning is the most accessible strategy. Our approach is designed for this realistic setting—providing calibration improvements without requiring access to earlier training stages.
> We have revised the paper to make this point clear and more explicit to readers.
>
> In the scenario where the practitioner is able to go through the full training pipeline, this is also not an issue. We provide our answer below.
>
> >Q2. What is the overall post-training pipeline? Justify the long post-training pipeline.
>
> A. CFT is a post-hoc fine-tuning step applied after RLHF, and we will clarify this in the revised manuscript. Following  our previous response, in scenarios where developers will go through the full training pipeline, both replacing RLHF and post-hoc fine-tuning are valid options—what matters most is which method yields better performance.
>
> Methods that aim to replace RLHF—such as PPO-C in [2]—are promising, but they are evaluated under different experimental settings, making a direct comparison difficult within a short timeframe. In our work, we demonstrate the effectiveness of our approach by outperforming the strong baseline of TS. We believe our method is robust and competitive with RLHF-replacement approaches.
>
> >Q3. It is an unfair comparison w/wo domain knowledge.
>
> A. We believe that the comparison remains appropriate for several reasons.
>
> First, regarding ECE, the concern about fairness seems to stem from intuition rooted in more standard metrics like accuracy, where additional domain knowledge can indeed lead to higher performance. However, ECE measures the discrepancy between confidence and accuracy, domain knowledge does not inherently bias ECE comparisons.
>
> Second, we have included experiments on out-domain tasks to further mitigate any potential effects from domain-specific knowledge.
>
> Finally, the ultimate goal of training language models is to obtain models that perform well in real-world settings. If incorporating additional knowledge leads to better-calibrated models, it should be considered a strength rather than a limitation.
>
> >Q4. It is unclear … hard to tell when people should use methods in Sec 5.1 and when to use methods in Sec 5.2.
>
> A: Before addressing it directly, we would like to first clarify a potential misunderstanding regarding Section 5.
>
> In Theorem 4.6, we prove that ECE ≤ TCE, which implies that minimizing TCE is sufficient for achieving low ECE. This reduces the goal of obtaining a well-calibrated model to finding one that is close to the target probabilistic model, formulated as the following constrained optimization problem:
> $$\max \text{ACC}(\pi) \quad \text{s.t.} \quad \text{ECE}(\pi) = 0.$$
>
> In Section 5, we focus on how to approximate both accuracy and ECE in the context of LLMs—covered in Sections 5.1 and 5.2, respectively.
>
> Now, returning to your question: Sections 5.1 and 5.2 do not present two separate methods, but rather provide a breakdown of our final algorithm.
>
> >Q5. I think the authors should not restrict to only four possible answers (only four-answer setting (A,B,C,D) is allowed), as that is not practical in real applications.
>
> A. We chose to use the four-option format (A, B, C, D) for concreteness and readability, aiming to reduce the burden of mathematical notation and make the presentation more accessible to experimental researchers. This is explicitly stated in our paper (lines 113–114): “While we consider four options in our analysis for concreteness, the framework naturally extends to any number of alternatives.” We will further revise the text to make this generality clearer.
>
> >Q6. The original preference alignment ability.
>
> In addition to the win rate results presented in our original submission, we have conducted three more experiments related to preference alignment: CFT vs. DPO, AlpacaEval, and Arena-Hard. Due to space constraints, we refer to our response to Q4 of QE3c and the end of our response to USq4 for the detailed results.
>
> >Q7. Related work, suggestions on explaining the two regimes at first and other writings.
>
> A. We have included the 4 mentioned papers in the revised version of the paper and have revised the paper accordingly.

---

> > ### Comment · Reviewer_hpMa · 2025-04-02
> >
> > The reviewers have addressed my concerns and I have raised my score. Please make sure to include the justification of your post training pipeline in the revised version.

---

> > > ### Author Response · Authors · 2025-04-04
> > >
> > > Thank you for the response. We'll make sure to clearly justify the post-training pipeline in the revised version.

---

### Official Review · Reviewer_USq4 · 2025-03-11

**Overall Recommendation:** 4

**Summary:**

This paper addresses poor calibration in Large Language Models (LLMs) after preference alignment procedures like RLHF and DPO. The authors identify that preference-aligned LLMs exhibit overconfidence due to "preference collapse," where models excessively favor certain responses regardless of their correctness. They develop a theoretical framework distinguishing between "calibratable" and "non-calibratable" regimes based on model accuracy thresholds, and propose two solutions: (1) Calibration-aware Fine-Tuning (CFT) for models in the calibratable regime, which restores calibration without compromising performance, and (2) Regularized CFT (RCFT) for the non-calibratable regime, which uses EM-algorithm-based ECE regularization to balance calibration and accuracy. Experiments across four models show their methods reduce Expected Calibration Error from 14-20% to 2-7% while maintaining or improving language capabilities and alignment with human preferences.

**Claims And Evidence:**

The paper's claims are generally supported by evidence, with experimental results clearly demonstrating ECE reductions from 14-20% to 2-7% across models.  The connection between preference collapse and poor calibration shows correlation but not definitive causation. Win rate metrics support their claim about maintaining alignment capabilities, but this represents just one dimension of alignment quality. While they demonstrate cross-domain generalization, more diverse evaluation scenarios would strengthen their broader applicability claims.

**Essential References Not Discussed:**

No.

**Experimental Designs Or Analyses:**

The experimental design is generally sound but has several limitations. While they use appropriate models (four different architectures with both DPO/RLHF alignment), metrics (ECE, accuracy, win rate), and testing conditions (in-domain and cross-domain), the analysis lacks statistical significance testing and confidence intervals. The ablation studies are minimal, only comparing LSFT2 vs LECE in the appendix. The regularization parameter λ is fixed at 1 without sensitivity analysis, and the win rate metric only captures binary preference alignment rather than nuanced quality dimensions.

**Methods And Evaluation Criteria:**

The calibration-aware fine-tuning methods directly target the identified overconfidence issue through targeted loss functions, which is sensible given the problem definition. Their use of multiple-choice QA datasets is appropriate since they provide clear ground truth for calculating calibration metrics. The confidence and classwise ECE metrics are standard for calibration assessment, making their evaluation protocol methodologically sound.

Their evaluation balances both in-domain and out-domain generalization, tests multiple model architectures aligned with different methods (DPO and RLHF), and importantly, measures win rate on preference pairs to verify alignment preservation. This comprehensive approach addresses the key concern that improving calibration might compromise alignment quality.

One limitation is that their evaluation focuses primarily on multiple-choice settings rather than free-form text generation, which would provide a more complete picture of calibration in real-world LLM deployments.

**Other Comments Or Suggestions:**

No.

**Other Strengths And Weaknesses:**

See comments above.

**Questions For Authors:**

1: How might your calibration methods be adapted for cases where we only have black-box API access to LLMs without fine-tuning capabilities?

2: Have you considered how quantization or other efficiency techniques might impact calibration properties in aligned models?

**Relation To Broader Scientific Literature:**

This work extends prior work on LLM calibration by Jiang et al. (2021), Xiao et al. (2022), and Chen et al. (2022) who identified miscalibration issues in LLMs, but specifically addresses the previously unexamined problem of how preference alignment techniques (RLHF/DPO) impact calibration.

**Theoretical Claims:**

I verified the paper's theoretical proofs, focusing on:

Proposition 4.1: Correctly proves that probabilistic generative models achieve zero ECE by definition, as predicted probabilities match observed frequencies.
Theorems 4.4 and 4.5 (Upper/Lower Bounds of TCE): The bounds are correctly established by constructing appropriate examples, but the constant C in the lower bound lacks specific derivation, making practical application less clear.
Theorem 4.6 (Upper bound for ECE): This uses triangle inequality to establish that classwise ECE is bounded by TCE, which is mathematically valid.

The calibratable/non-calibratable regime distinction follows logically from these bounds, though the paper simplifies this conceptually when moving to practical implementation. The EM-algorithm for probability estimation is theoretically sound, though the proof for convergence is not provided.

---

> ### Author Rebuttal · Authors · 2025-04-01
>
> We thank reviewer USq4 for the insightful comments and questions. Below we provide our responses to the questions.
>
> >Q1. One limitation is that their evaluation focuses primarily on multiple-choice settings rather than free-form text generation, which would provide a more complete picture of calibration in real-world LLM deployments.
>
> A. Thank you for the comment. We agree that calibration in free-form generation is important. We focus on multiple-choice settings as they provide a controlled and quantifiable evaluation of calibration. Extending our method to free-form generation is an exciting direction, and we will discuss it as future work.
>
> >Q2. the constant C in the lower bound lacks specific derivation,
>
> A. Thank you for pointing this out. The constant C in the lower bound arises from solving a min-max optimization problem, which makes it difficult to obtain a closed-form expression. While we do not derive an explicit value for C, we will clarify its origin and theoretical role in the revised manuscript. Characterizing or approximating C more precisely is an interesting direction for future work.
>
> >Q3. The EM-algorithm for probability estimation is theoretically sound, though the proof for convergence is not provided.
>
> A. Thank you very much for the question. When the optimization is performed over probability distributions (rather than neural network parameters), the convergence of our EM algorithm follows from standard EM theory, and we take this as given rather than presenting it as a contribution of our work. However, when optimizing over neural network parameters, convergence is no longer guaranteed due to the non-convexity and complexity of the underlying function space.
> We will include a discussion on the convergence properties of the algorithm in the revised manuscript to clarify these distinctions.
>
> >Q4. The experimental design is generally sound but has several limitations. While they use appropriate models (four different architectures with both DPO/RLHF alignment), metrics (ECE, accuracy, win rate), and testing conditions (in-domain and cross-domain), the analysis lacks statistical significance testing and confidence intervals. The ablation studies are minimal, only comparing LSFT2 vs LECE in the appendix. The regularization parameter λ is fixed at 1 without sensitivity analysis, and the win rate metric only captures binary preference alignment rather than nuanced quality dimensions.
>
> A. Thank you very much for the helpful comments and suggestions. We have now conducted a more comprehensive ablation study on the hyperparameter λ to further support our analysis. The results of this study have been incorporated into the revised manuscript.
> |$\lambda$|0||| 0.4|||1|||1.8|||ECE_only|||
> |-|-|-|-|-|-|-|-|-|-|-|-|-|-|-|-|
> |Metric|ECE|cwECE|Acc|ECE|cwECE|Acc|ECE|cwECE|Acc|ECE|cwECE|Acc|ECE|cwECE|Acc|
> |Llama-3.1-8B-Tulu| 0.0883 | 0.0808 | 0.8964 | 0.1535 | 0.1014 | 0.8409 | 0.0897 | 0.0771 | 0.8341 | 0.0178 | 0.0106 | 0.4366 | 0.0002 | 0.0081|0.2475|
> |Vicuna-7B| 0.1219 | 0.0774 | 0.8322 | 0.1620 | 0.0991 | 0.7315 | 0.0474 | 0.0459 | 0.6015 | 0.1052 | 0.0799 | 0.3877 | 0.0130 | 0.0270 | 0.2290 |
> |Olmo2-7B| 0.1003 | 0.0992 | 0.8846 | 0.1771 | 0.1008 | 0.8427 | 0.0989 | 0.0806 | 0.8510 | 0.0038 | 0.0113 | 0.4901 | 0.0030 | 0.0043 | 0.2765|
> |Mistral-7B| 0.0976 | 0.0785 | 0.9091 | 0.1316 | 0.0733 | 0.8085 | 0.0979 | 0.0877 | 0.8297 | 0.0366 | 0.0617 | 0.4217 | 0.0021 | 0.0108 | 0.2670 |
>
> >Q5. How might your calibration methods be adapted for cases where we only have black-box API access to LLMs without fine-tuning capabilities?
>
> A. Thank you for the question. Our approach relies on fine-tuning the model, and is therefore not applicable to black-box APIs. Adapting our method to black-box models would go beyond a simple extension—it would effectively require developing an entirely new approach. For improving calibration in black-box settings, techniques such as prompt engineering or encouraging the model to explicitly express its confidence are more suitable. We will include this discussion in the revised version of the paper.
>
> >Q6. Have you considered how quantization or other efficiency techniques might impact calibration properties in aligned models?
>
> A. Thank you very much—this is a great question. Previously, we did not explore the impact of quantization or other efficiency techniques on calibration. These methods can potentially alter the model's confidence estimates and thus affect calibration quality. We agree this is an important and practical direction, especially for deployment scenarios, and we will include it in our discussion of future work.
> ___
> To all reviewers: Additional experiments of alignment ability. Due to limited time, only 2 models for Arena-hard.
> |Alpaca-Eval|DPO| CFT| RCFT |
> |-|-|-|-|
> |  Llama-3.1-8B-Tulu |  21.4|22.6|19.6|
> | Vicuna-7B| 2.6|2.6|3.6|
> | Olmo2-7B | 24.2|22.9|23.1|
> | Mistral-7B| 26.0|26.8|25.2|
> |Arena-hard||||
> | Olmo2-7B|19.4|19.2|20.2|
> | Mistral-7B| 18.9|18.3|18.0|

---

### Official Review · Reviewer_Ak1F · 2025-03-12

**Overall Recommendation:** 2

**Summary:**

This paper addresses the calibration issue in aligned large language models (LLMs) and proposes a calibration-aware fine-tuning approach to restore proper uncertainty quantification in these models. The motivation stems from the observation that alignment techniques can distort model confidence, leading to miscalibrated probabilities in downstream tasks.
The proposed method introduces a fine-tuning procedure that explicitly optimizes for calibration metrics while preserving the alignment properties of the LLM. The approach is designed to be agnostic to the underlying alignment strategy, making it adaptable to different LLM architectures and training paradigms. The authors provide theoretical insights into how alignment affects calibration and demonstrate the effectiveness of their method through extensive empirical evaluations on multiple benchmarks.

## update after rebuttal

First, I would like to apologize for not being able to respond directly to the authors during the discussion phase, due to limitations of the review system.
Instead, I am using this update to clearly state my current position.

The authors have provided a strong and effective rebuttal that addresses many of my original concerns.
In particular, I sincerely appreciate their effort in conducting additional experiments based on theoretically justified bin sizes—especially considering how challenging such LLM experiments must be.

That said, I still have two remaining concerns:
- **Presentation clarity:** I believe the paper does not sufficiently distinguish between the ECE used for evaluation and the ECE used in the objective function. Given that multiple definitions of “ECE” appear throughout the paper, I found it somewhat confusing—even as someone who works regularly with calibration metrics. Improving this distinction would greatly help with readability.

- **Distinction between fine-tuning and standard classification tasks:** While I agree with the authors’ explanation regarding how labels are handled differently in the objective, I still find it unclear why methods from classification settings would not be applicable here.After all, the goal—minimizing ECE—is common to both this work and prior approaches in classification calibration. Is the difference simply in how probabilities are computed with respect to the labels? (It is possible that I am still misunderstanding this point.)I believe it is crucial to clearly articulate the distinction between the two approaches.If that proves difficult, an alternative could be to empirically compare the proposed method with existing calibration regularization techniques developed for classification—such as smooth CE or other differentiable calibration metrics—applied in this context. This would be a simple yet effective way to differentiate your method.

I would like to raise these points for discussion among the reviewer panel. For now, I will keep my original score, but depending on the outcome of the discussion, I may consider increasing it.

Lastly, I would like to emphasize that I find this to be a very interesting and promising paper.

**Claims And Evidence:**

### Claim 1 (Explaining why alignment affects calibration):

The paper provides an intuitive argument that preference alignment alters model confidence by enforcing human-desired responses, potentially distorting probability estimates. While it is not entirely clear which part of the  results explicitly show the critical answer for "why preference alignment affects calibration", the fine-tuning approach naturally introduces a trade-off between accuracy and calibration error, which is well-motivated. The design of the objective function balances these two aspects in a reasonable manner, making the proposed method a natural approach to addressing miscalibration in aligned LLMs.


### Claim 2 (Proposed method mitigates calibration issues):
While the paper focuses on the LLM fine-tuning, the underlying problem is fundamentally equivalent to the long-standing question in the machine learning community: how to balance classification accuracy and calibration in predictive models.
Although the specific task setting is different, the challenge addressed here is closely related to classical studies in calibration-aware learning.

The proposed fine-tuning approach somewhat reduces calibration error in experiments. However, the justification for why this method is the best solution is weak.
- Why is fine-tuning with a calibration objective superior to alternative methods such as temperature scaling?
- Good predictive accuracy and calibration simultaneously have been widely studied in general machine learning classification problems (please see the following references). Beyond differences in objectives and tasks, what is the fundamental distinction between these approaches and the proposed method? Why is the proposed method preferable to extending those studies? Could some of the methods proposed in the following papers be incorporated into the fine-tune objective of the LLM without too much computational complexity?
    - [Kumar et al., ICML2018](https://proceedings.mlr.press/v80/kumar18a/kumar18a.pdf#:~:text=Trainable%20Calibration%20Measures%20For%20Neural,so%20compro%02mise%20the%20many%20legitimately)
    - [Krishnan et al, NeurIPS2020](https://arxiv.org/pdf/2012.07923)
    - [Karandikar et al., NeurIPS2021](https://arxiv.org/pdf/2108.00106)
    - [Popordanoska et al., NeurIPS2022](https://proceedings.neurips.cc/paper_files/paper/2022/file/33d6e648ee4fb24acec3a4bbcd4f001e-Paper-Conference.pdf#:~:text=We%20propose%20a%20tractable%2C%20differentiable%2C,ECEKDE%20scales%20well)


### Claim 3 (Empirical results demonstrate effectiveness without degrading model performance):

In p.6, the Evaluation Metric section defines ECE as a nonparametric estimator based on binning. Recent studies have shown that this estimator suffers from significant estimation bias (e.g., [Futami & Fujisawa, NeurIPS2024](https://proceedings.neurips.cc/paper_files/paper/2024/file/9961e42624a6c083279303767c73269d-Paper-Conference.pdf), for binary classification; [Fujisawa & Futami, 2024](https://arxiv.org/pdf/2406.06227), for multiclass classification).

According to these works, binning-based ECE has a slow convergence rate of $O(1/n^{1/3})$ and introduces a significant bias. The optimal number of bins to minimize this bias in an upper-bound sense is $O(n^{1/3})$, but this paper fixes the number of bins to $10$, which likely results in substantial estimation bias.

Given this, the numerical results presented in the paper should be interpreted with caution, as the validity of the estimated calibration error is questionable.

**Essential References Not Discussed:**

To properly position the contribution of this work and ensure a fair evaluation of calibration performance, it is essential to reference and discuss the prior research on calibration-aware learning and bias in ECE estimation that has been highlighted in this review.

- The paper should cite and discuss prior works on calibration-aware learning in classification models (e.g., [Kumar et al., ICML2018](https://proceedings.mlr.press/v80/kumar18a/kumar18a.pdf#:~:text=Trainable%20Calibration%20Measures%20For%20Neural,so%20compro%02mise%20the%20many%20legitimately), [Krishnan et al, NeurIPS2020](https://arxiv.org/pdf/2012.07923), [Karandikar et al., NeurIPS2021](https://arxiv.org/pdf/2108.00106), [Popordanoska et al., NeurIPS2022](https://proceedings.neurips.cc/paper_files/paper/2022/file/33d6e648ee4fb24acec3a4bbcd4f001e-Paper-Conference.pdf#:~:text=We%20propose%20a%20tractable%2C%20differentiable%2C,ECEKDE%20scales%20well)), which are highly relevant to the methodology of this paper.

- The validity of ECE-based evaluation should be reconsidered, given recent studies showing significant bias in binning-based ECE estimators ([Gruber et al., NeurIPS2022](https://arxiv.org/pdf/2203.07835), [Sun et al., NeurIPS2023](https://arxiv.org/pdf/2305.10886), [Gupta et al., ICML2021](https://arxiv.org/pdf/2105.04656), [Gupta et al., NeurIPS2020](https://arxiv.org/pdf/2006.10564), [Futami & Fujisawa, NeurIPS2024](https://proceedings.neurips.cc/paper_files/paper/2024/file/9961e42624a6c083279303767c73269d-Paper-Conference.pdf), [Fujisawa & Futami, 2024](https://arxiv.org/pdf/2406.06227)). If ECE is used as a primary evaluation metric, proper adjustments (e.g., optimal bin selection or alternative estimators) should be considered.

**Experimental Designs Or Analyses:**

As I mentioned above, the evaluation relies on ECE with binning, which introduces significant estimation bias.
As shown in [Futami & Fujisawa, NeurIPS2024](https://proceedings.neurips.cc/paper_files/paper/2024/file/9961e42624a6c083279303767c73269d-Paper-Conference.pdf) and [Fujisawa & Futami, 2024](https://arxiv.org/pdf/2406.06227), binning-based ECE has a slow convergence rate and can significantly over- or under-estimate calibration error.
The choice of $10$ bins likely introduces substantial bias, affecting the reliability of reported calibration improvements.
Alternative or bias-corrected calibration metrics should be considered to ensure more robust evaluation.

**Methods And Evaluation Criteria:**

Many of the concerns regarding the methods and evaluation criteria have already been discussed in the above section. To briefly summarize:

- While the paper focuses on LLM fine-tuning, the underlying problem is fundamentally equivalent to the long-standing question in the machine learning community: how to balance classification accuracy and calibration in predictive models.
Although the specific task setting is different, the challenge addressed here is closely related to classical studies in calibration-aware learning. The justification for why this method is the best solution remains weak:
    - Why is fine-tuning with a calibration objective superior to alternative methods such as temperature scaling?
    - Could existing methods from classification calibration research (e.g., those by [Kumar et al., ICML2018](https://proceedings.mlr.press/v80/kumar18a/kumar18a.pdf#:~:text=Trainable%20Calibration%20Measures%20For%20Neural,so%20compro%02mise%20the%20many%20legitimately), [Krishnan et al, NeurIPS2020](https://arxiv.org/pdf/2012.07923), [Karandikar et al., NeurIPS2021](https://arxiv.org/pdf/2108.00106), [Popordanoska et al., NeurIPS2022](https://proceedings.neurips.cc/paper_files/paper/2022/file/33d6e648ee4fb24acec3a4bbcd4f001e-Paper-Conference.pdf#:~:text=We%20propose%20a%20tractable%2C%20differentiable%2C,ECEKDE%20scales%20well) be incorporated into the fine-tuning objective without excessive computational complexity?
- The empirical evaluation has methodological concerns, particularly regarding the estimation bias in ECE due to fixed binning (see [Futami & Fujisawa, NeurIPS2024](https://proceedings.neurips.cc/paper_files/paper/2024/file/9961e42624a6c083279303767c73269d-Paper-Conference.pdf), [Fujisawa & Futami, 2024](https://arxiv.org/pdf/2406.06227)). The choice of 10 bins likely introduces substantial estimation bias, affecting the validity of the reported calibration improvements.

**Other Comments Or Suggestions:**

- In Definition 4.2: Is it needed to provide the definition of $\mathrm{ECE}(\pi_{\theta})$?

- In Appendix A.2, line 575: "Expected Calibration Error (ECE) Naeini et al. (2015),...": I think you should use the bibtex-style citation here for [Naeini et al. (2015)].

- In Appendix A.2, line 576: (typo?) "Mlticlass-ECE..." --> "Multiclass-ECE..."

- In Figures 7 and 8 (p.20-21): Why is the plot of (g) for "Prob. of Option A"? I think you should show the plot for "Prob. of Option B" instead.

**Other Strengths And Weaknesses:**

### Strengths
- Addresses an important issue in LLM alignment: The paper tackles the important problem of calibration degradation due to alignment, which is highly relevant given the widespread use of LLMs in real-world applications.
- Proposes a well-motivated fine-tuning objective: The method explicitly balances accuracy and calibration, making it a conceptually reasonable approach to mitigating miscalibration.
- Provides a theoretical framework: While certain connections could be clarified further, the paper presents a structured theoretical analysis, which helps to ground the empirical observations.
- Extensive empirical evaluation across multiple LLMs: The paper tests its method on various LLM architectures, demonstrating general applicability.

### Weaknesses
Lack of discussion on the limitations of the proposed method: The paper does not provide a sufficient discussion on when and where the proposed method is expected to perform well or poorly. For example:

- What are the computational costs associated with this fine-tuning method, and how do they compare to other calibration techniques?

- Are there trade-offs in terms of sample efficiency, convergence rate, or stability? Providing some empirical evaluation regarding this topic would help clarify the practical applicability of the proposed method.

**Questions For Authors:**

The following questions are critical for improving the clarity and contribution of the paper.
If these issues are appropriately addressed, I will consider raising my overall score.

- Comparison with Alternative Calibration Approaches:
    - Why is fine-tuning with a calibration objective superior to existing post-hoc calibration methods such as temperature scaling?
    - What is the relationship and significant difference between this paper and calibration-aware training in classification models, such as those by [Kumar et al., ICML2018](https://proceedings.mlr.press/v80/kumar18a/kumar18a.pdf#:~:text=Trainable%20Calibration%20Measures%20For%20Neural,so%20compro%02mise%20the%20many%20legitimately), [Krishnan et al, NeurIPS2020](https://arxiv.org/pdf/2012.07923), [Karandikar et al., NeurIPS2021](https://arxiv.org/pdf/2108.00106), and [Popordanoska et al., NeurIPS2022](https://proceedings.neurips.cc/paper_files/paper/2022/file/33d6e648ee4fb24acec3a4bbcd4f001e-Paper-Conference.pdf#:~:text=We%20propose%20a%20tractable%2C%20differentiable%2C,ECEKDE%20scales%20well)?

- Evaluation Metrics and ECE Bias:
    - The paper uses binning-based ECE, which is known to suffer from significant estimation bias (e.g., [Futami & Fujisawa, NeurIPS2024](https://proceedings.neurips.cc/paper_files/paper/2024/file/9961e42624a6c083279303767c73269d-Paper-Conference.pdf), [Fujisawa & Futami, 2024](https://arxiv.org/pdf/2406.06227))).
    - Could you provide revised experimental results using an optimal bin size setting as suggested by these studies? I think this is crucial to confirm whether the reported empirical results are convincing or not.
    - Does the paper employ uniform-mass binning (UMB) or uniform-width binning? Since the choice of binning strategy significantly impacts bias and variance, providing clarification would be valuable.

- Limitations of the Proposed Method:

    - In which scenarios is the proposed method expected to perform well, and in which cases might it fail?

    - Are there certain types of LLM architectures, alignment techniques, or dataset conditions where this approach is less effective?

    - What are the computational costs of this fine-tuning method? How do they compare to other calibration techniques? Is the ECE term in the proposed objective function differentiable?

**Relation To Broader Scientific Literature:**

This work is closely related to prior research in calibration-aware learning in classification models, where calibration metrics are incorporated as regularization terms in the objective function to jointly optimize predictive accuracy and calibration performance. Although this paper focuses on fine-tuning LLMs, the core problem it addresses is fundamentally equivalent to the long-standing challenge in classification models:　balancing accuracy and calibration in predictive learning.

Several prior studies have explored this challenge:

- [Kumar et al., ICML2018](https://proceedings.mlr.press/v80/kumar18a/kumar18a.pdf#:~:text=Trainable%20Calibration%20Measures%20For%20Neural,so%20compro%02mise%20the%20many%20legitimately): One of the earliest works to directly integrate Expected Calibration Error (ECE) optimization into model training. They propose the Maximum Mean Calibration Error (MMCE), a new metric based on RKHS kernels, which is added as a regularization term to the cross-entropy loss.

- [Krishnan et al, NeurIPS2020](https://arxiv.org/pdf/2012.07923): Introduce a differentiable loss that explicitly penalizes the discrepancy between model confidence and actual correctness, enabling direct optimization of calibration error.

- [Karandikar et al., NeurIPS2021](https://arxiv.org/pdf/2108.00106): Develop a differentiable calibration loss by continuous relaxation of histogram binning for ECE computation. Their method, when incorporated into training, significantly reduces ECE for individual models.

- [Popordanoska et al., NeurIPS2022](https://proceedings.neurips.cc/paper_files/paper/2022/file/33d6e648ee4fb24acec3a4bbcd4f001e-Paper-Conference.pdf#:~:text=We%20propose%20a%20tractable%2C%20differentiable%2C,ECEKDE%20scales%20well): Propose a Kernel Density Estimation-based ECE estimator, making Canonical Calibration Error (an extension of ECE) differentiable. This allows it to be integrated into the training objective, achieving a favorable trade-off between accuracy and calibration performance.

Beyond calibration-aware training methods, the validity of ECE-based uncertainty estimation itself has been questioned in recent studies.
As highlighted by [Futami & Fujisawa, NeurIPS2024](https://proceedings.neurips.cc/paper_files/paper/2024/file/9961e42624a6c083279303767c73269d-Paper-Conference.pdf) and [Fujisawa & Futami, 2024](https://arxiv.org/pdf/2406.06227) and [Fujisawa & Futami, 2024](https://arxiv.org/pdf/2406.06227), binning-based ECE estimators suffer from significant estimation bias and slow convergence rates.

Additionally, some studies have provided theoretical evaluations of bias in ECE estimators based on uniform-mass binning (UMB)([Gupta et al., ICML2021](https://arxiv.org/pdf/2105.04656), [Gupta et al., NeurIPS2020](https://arxiv.org/pdf/2006.10564)).
There are some other related studies ([Gruber et al., NeurIPS2022](https://arxiv.org/pdf/2203.07835), [Sun et al., NeurIPS2023](https://arxiv.org/pdf/2305.10886)).
It is unclear whether this paper uses UMB or uniform-width binning, but the choice of binning strategy significantly impacts bias and variance in calibration error estimation. Given these findings, it is important to acknowledge the potential limitations of ECE-based evaluation and consider methods that minimize bias, such as properly setting the number of bins or using alternative bias-corrected estimators. Including such discussions would strengthen the empirical robustness of this work.

**Theoretical Claims:**

I have reviewed the proofs and found no obvious errors; the mathematical derivations appear to be correct.

Assuming that the proofs are entirely correct, the claims derived from the theorems are reasonable in themselves. However, it is not clear that these results provide a direct answer to the question of "why preference alignment affects calibration." Instead, the theoretical findings seem to demonstrate the trade-off between accuracy and calibration, a well-known issue that has been widely discussed in the general machine learning community.

From a scientific perspective, a more appropriate claim may be that the paper formalizes this accuracy-calibration trade-off within the context of fine-tuning LLMs, rather than providing a direct theoretical justification for the miscalibration induced by preference alignment.

---

> ### Author Rebuttal · Authors · 2025-04-01
>
> We thank reviewer Ak1F for the insightful comments and questions. Below we provide our responses to the questions.
>
> >Q1. Why fine-tuning with a calibration objective superior to post-hoc calibration methods e.g. TS?
>
> A. While temperature scaling (TS) is a strong baseline, our method's superiority stems from addressing fundamental limitations of post-hoc calibration in LLMs:
>
> 1. Our approach explicitly minimizes the discrepancy between accuracy and confidence—the definition of ECE—rather than relying on a single scaling parameter. This direct optimization improves calibration while maintaining or enhancing performance. This aligns with previous work on training classifiers with calibration objectives, such as the AvUC loss in [Krishnan et al](https://arxiv.org/pdf/2012.07923) and the PAC-Bayes-based objective in [Fujisawa & Futami, 2024](https://arxiv.org/pdf/2406.06227), which demonstrated the superiority of optimization-based strategies over post-hoc methods.
>
> 2. Our approach generalizes better to unseen data and distribution shifts compared to TS, which risks overfitting to specific validation sets. Table 2 clearly demonstrates this advantage in out-domain scenarios—for example, with Olmo2-7B, our CFT method reduces out-domain cw-ECE to 0.0637 (vs. TS's 0.1196) while improving accuracy to 0.7085 (vs. DPO's 0.6635). Distribution shifts are common in language tasks, and TS is insufficient to handle such variations.
>
> We will include these points in our revised manuscript.
>
> >Q2. Relationship between this paper and calibration-aware training in classification models?
>
> A. The referenced works propose various calibration-aware training objectives for classification models, such as MMCE, AvUC, S-AvUC, and SB-ECE. Some of these are designed as accuracy–calibration trade-off objectives.
>
> Our approach shares this general principle, as it can also be viewed as an accuracy–calibration trade-off method. The key difference is that our accuracy and calibration objectives are specifically designed for generative models and LLMs, rather than classification models.
>
> For the accuracy objective, we use the SFT loss on next-token prediction. This loss not only promotes accuracy but also supports instruction following, knowledge grounding, and coherent text generation—all essential to LLMs.
>
> For the ECE objective, we formulate calibration as a generative modeling problem and apply an EM algorithm to optimize it accordingly.
>
> In summary, our method is tailored for LLMs, which represents a significant departure in both design and application.
>
> >Q3. Evaluation Metrics and ECE Bias and optimal bin size.
>
> A. Thank you for the helpful suggestion. The two referenced papers on ECE bias and optimal bin size are indeed insightful.
>
> In our work, we use the UWB strategy, and we will make this choice explicit in the revised manuscript.
>
> According to the referenced work, the optimal bin size for the multiclass setting scales as $O(n^{1/3})$. However, to apply this practically, one needs the exact constant rather than just the asymptotic rate. Upon further examination, we found that the optimal bin size contains a Lipshitz constant of the model, in a rate of $(1 + L)^{2/3} n^{1/3}.$
>
> In practice, estimating the Lipschitz constant for transformers is challenging, as it is known to be potentially very large and difficult to compute reliably. Nonetheless, we can still make use of the theoretical rate $n^{1/3}$ as in the two mentioned papers. In our experiments, the sample size is 3,000, which implies an optimal bin size on the order of O(14.4). The bin size of 10 used in our paper is reasonably close to this rate, suggesting that our choice is consistent with the theoretical guidance.
>
> >Q4. In which cases the proposed methods perform well and fail?
>
> A. Our approach is specifically designed and performs well for LLMs which are originally well-calibrated but become poorly calibrated after preference alignment. In other scenarios—such as (1) traditional classification tasks, or (2) cases where the LLM is not well-calibrated to begin with—our method may not be as effective.
>
> >Q5. Certain types of LLM architectures, alignment techniques ... where this approach is less effective?
>
> A. In our experiments, we evaluate the method across different architectures, alignment techniques, and datasets to demonstrate its generality. However, due to computational constraints, we did not experiment with large-scale models such as 70B. While the effectiveness may vary at that scale, we believe our method is conceptually scalable and can be applied to larger models with appropriate resources.
>
> >Q6. Computational costs? Compare to other techniques? ECE term differentiable?
>
> A. On two A100 40GB GPUs, our approach takes approximately 1.5 hours to run for 5 epochs. The training time is comparable to label smoothing, but significantly slower than temperature scaling, as the latter does not require fine-tuning model weights. In addition, Our ECE term is differentiable.

---

> > ### Comment · Reviewer_Ak1F · 2025-04-03
> >
> > Thank you very much for your thoughtful and respectful rebuttal. I truly appreciate the care you have taken in addressing each point in detail.
> >
> > ## Regarding Q1:
> > Thank you for your detailed explanation. I would strongly encourage incorporating this discussion into the main text. Fujisawa & Futami (2024) also discuss how Temperature Scaling (TS) can sometimes achieve good recalibration performance depending on the order of the optimal bin size used for ECE estimation. In light of this, and in conjunction with the discussion in Q3, I would recommend re-evaluating your performance results to ensure validity of your proposed method.
> >
> > ## Regarding Q2:
> > I understand that your setting—fine-tuning with preference data—is different from standard classification, and that your proposed objective is tailored for LLMs. The method is indeed interesting. That said, when viewed from a broader perspective, your formulation seems conceptually equivalent to a widely discussed approach in the literature that minimizes an objective of the form: (accuracy loss) + $\lambda$ × (calibration regularization).
> > Therefore, I believe it would be helpful to more deeply explore and clarify your argument that classification and fine-tuning are fundamentally distinct. Are they truly different in essence? In basic preference fine-tuning settings, models are often trained to behave in a way that aligns with binary labels indicating whether outputs are preferred or not. Despite differences in reward function design or likelihood formulation, the underlying mechanism often involves computing binary classification probabilities via a logistic function. A deeper clarification of how your method fundamentally differs from classification-based regularization would help readers better appreciate your contribution.
> >
> > ## Regarding Q3:
> > You are absolutely right about the Lipschitz constant. It makes sense to choose it based on the order of $\mathcal{O}(n^{1/3})$ in practice. That said, in my personal experience, the actual ECE value can vary significantly depending on whether the number of bins is 10 or 14. I understand that re-running LLM experiments can be computationally intensive, but re-evaluating the performance—including that of TS—under these settings would further substantiate your claims. Ideally, showing the updated numbers would be most helpful.
> >
> > ## Regarding Q4:
> > Thank you for proposing an intriguing hypothesis. Could you consider expanding this point into a discussion of the potential limitations of your method? Clearly stating in which cases the method is effective and where it may face challenges would not only enhance the practical value of your work but also help guide future research.
> >
> > ## Regarding Q5:
> > I appreciate your response regarding computational constraints. If possible, I would suggest including even a theoretical discussion on computational complexity or convergence behavior. (Although I may be mistaken, would the E-step, for example, have a complexity of $\mathcal{O}(N × M^{2})$?)
> >
> > ## Regarding Q6:
> > I may have overlooked something, but I was curious—how is the objective based on nonparametric estimation through binning differentiable? Or is it that the ECE used for evaluation differs from the one used in your objective? Section 3 introduces cw-ECE and conf-ECE, whereas Section 4 refers simply to “ECE,” which left me slightly confused. It would be very helpful if you could clarify this distinction.

---

> > > ### Author Response · Authors · 2025-04-07
> > >
> > > Thanks for the responses, and we are sorry for the delayed reply, as conducting additional experiments required some time.
> > > >Q1 & 3.
> > >
> > > Thank you for the suggestion. We will incorporate the discussion into the main text.
> > >
> > > Following the the referenced paper, we have chosen the bin size=14, and re-evaluated all four methods across architectures, types of ECE, in both in/out-domain settings.
> > >
> > > We observe that CFT and RCFT consistently restore the ECE of DPO models. The comparison between our approach and TS remains consistent with our original findings. In 5 out of 8 conf-ECE comparisons, CFT outperforms TS. Additionally, we find that the ECE values of TS and CFT are closer under the new binning choice, which partially supports the idea that this bin size reduces ECE bias.
> > >
> > > We will include a discussion on the bin size rate of the referenced paper and these updated experimental results, in our revised paper.
> > >
> > > ||Llama3.1 In|Out|Olmo2 In|Out|Vicuna In|Out|Mistral In|Out|
> > > |-|-|-|-|-|-|-|-|-|
> > > |DPO|0.1861/0.0988|0.1188/0.0657|0.1370/0.0773|0.0914/0.0630| 0.1418/0.0664|0.0888/0.0993|0.1979/0.1010|0.1346/0.1187|
> > > |TS|0.1158/0.0349|0.0559/0.0256|**0.0490**/0.0329| **0.0272**/0.0252|0.0377/0.0220|**0.0297**/0.0523|0.0771/0.0380|0.1093/0.0582|
> > > |CFT|**0.0441**/0.0418|**0.0520**/0.0344|0.0587/0.0376|0.0573/0.0356|**0.0216**/0.0295|0.0308/0.0516|**0.0602**/0.0207|**0.0511**/0.0601|
> > > |RCFT|0.1011/0.0783|0.0801/0.0525|0.0730/0.0365|0.0663/0.0512|0.0508/0.0397|0.0677/0.0552|0.0817/0.0457|0.0658/0.0506
> > >
> > > The results are reported in the format: conf-ECE / cw-ECE.
> > > >Q2.
> > >
> > > Let's clarify the distinction between preference fine-tuning and classification.
> > >
> > > **Objective.** As you mentioned, the objective of preference alignment can be viewed as a classification loss with two classes “winner” or “loser”. However, this is fundamentally different from a standard classification problem. To illustrate, consider a simplified setting with three responses: $r_1, r_2, r_3$, where human preferences are $r_1 > r_2 > r_3$. The preference alignment objective introduces the following pairwise comparisons into the loss: $(r_1, r_2)$, $(r_2, r_3)$, and $(r_1, r_3)$. In the first comparison, $r_2$ is labeled as the loser since $r_1$ is preferred. However, in the second comparison, $r_2$ is labeled as the winner because it is preferred over $r_3$. Thus, the same response ($r_2$) appears as both a winner and a loser depending on the comparison.
> > >
> > >
> > > This shows that there is no consistent global label for each response—labels are defined only in the context of pairwise comparisons. A single response can belong to both the winner and loser classes in different pairs. In contrast, standard classification tasks (such as MNIST digit recognition) involve a clear and consistent labeling scheme. Each image belongs to exactly one of the digit classes (0 through 9), and these labels are globally defined and mutually exclusive.
> > >
> > > **Evaluation of calibration.** In standard classification settings, calibration error is typically evaluated on the same task. In contrast, for LLMs, calibration is often evaluated on different tasks—such as multiple-choice questions with four options (four classes)—which are distinct from the preference alignment task (two classes).
> > >
> > > Therefore, while adding a calibration regularization term is common practice in standard classification settings, directly applying this approach to the preference alignment loss lacks principled justification in this context. In contrast, our approach—post-hoc fine-tuning after the alignment stage—is more intuitively and conceptually appropriate, since both the accuracy and calibration losses are designed with the downstream evaluation task in mind.
> > >
> > > >Q4.
> > >
> > > Sure, we will include the discussion of limitations, as outlined in our previous responses, in the revised version of the paper.
> > > >Q5:
> > >
> > > The total computational complexity is given by
> > > $$L(5n + M + n/B),$$
> > > where $L$ is the number of epochs and $B$ is the batch size.
> > >
> > > E-step. It requires n steps from the outer loop. We acknowledge that the inner loop, as currently written, may be misleading. It follows the standard EM-algorithm format but does not reflect the actual computational procedure. In practice, there is no need to iterate over all $M$ bins to determine bin membership for each sample.
> > >
> > > M-step.
> > > - $M$ steps to update the accuracies of the \(M\) bins.
> > > - $4n$ steps to update the four target probabilities for each samples.
> > > - $n/B$ gradient steps to update the model parameters.
> > >
> > > Regarding convergence, we will include a discussion on the standard convergence analysis of EM algorithms in the revised version of the paper.
> > > > Q6.
> > >
> > > The ECE in the objective and  for evaluation are different. In Section 5.2, we define the ECE loss as:
> > > $$
> > > L_{ECE}=D(p(x),conf_\pi(x)),
> > > $$
> > > where D is a differentiable divergence (we use the MSE). Regarding the binning process: $p(x)$ is fixed after binning and its gradient is detached. As a result, the overall loss is differentiable.

---

### Official Review · Reviewer_QE3c · 2025-03-14

**Overall Recommendation:** 3

**Summary:**

The paper tries to answer a well-known question: why an aligned model is not well-calibrated and how to fix it? Authors start with a probabilistic generative model and define TCE, and then derive an upper and lower bound of TCE. Then, depends on the accuracy of the current model, one can either get calibration without sacrificing accuracy or not. The authors state that, most cases are calibratable.

To recover the calibration of models, authors propose an EM algorithm to compute calibration loss. Together with SFT loss, the CFT get better calibration loss and accuracy.

**Claims And Evidence:**

* Upper and lower bound of TCE. Suppored by Thm 4.4-4.6
* Calibratable and Uncalibratable cases based on Accuracy. Supported by Thm 4.4-4.6.
* EM algorithms to recover calibration. Supported by experiment.

**Essential References Not Discussed:**

Some relevant methods are not  discussed:
[1] https://arxiv.org/abs/2102.09690
[2] https://arxiv.org/pdf/2405.20974

**Experimental Designs Or Analyses:**

The experiment is overall sound.

### Question and Weakness
* There is only one baseline (temp scaling); it would be better to compare with other baselines such as label smoothing, [1], etc.
* A better comparison of the Win rate is between CFT and DPO-generated responses since response qualities may decrease but still outperform the one being compared in the current setting.

[1] https://arxiv.org/abs/2102.09690

**Methods And Evaluation Criteria:**

Methods and evaluation makes sense.

**Other Comments Or Suggestions:**

## update after rebuttal

Since the authors addressed my concerns, I raised my score to 3.

**Other Strengths And Weaknesses:**

Other strengths:
* The paper is generally well-written.

Other weakness:
* See above

**Questions For Authors:**

N/A

**Relation To Broader Scientific Literature:**

The paper aligns a line of calibration work such as temperature scaling, Say-Self, etc. This work gives a theory analysis of calibration and a new EM calibration algorithm, which differs from prior works.

**Theoretical Claims:**

I did not carefully check the proof Thm4.4 - 4.6, but they seem correct.

### Question & Weakness
* One theory claims that both calibratable and un-calibratable cases exist. Can we draw any senses about when the model falls into each case from a theoretical perspect?
* The theory contribution is unclear. If my understanding is correct, the theory mainly states that there are two possible scenarios and does not directly relate to other parts of the paper, such as algorithms. The insight drawn from the theory is vague.

---

> ### Author Rebuttal · Authors · 2025-03-31
>
> We thank reviewer QE3c for the insightful comments and questions. Below we provide our responses to the questions.
>
> >Q1. Can we draw any senses about when the model falls into each case from a theoretical perspect?
>
> A. This is a great question. Determining which regime a model falls into ultimately reduces to understanding whether its accuracy exceeds a certain threshold. This, in turn, depends on two key factors: 1. What is the accuracy threshold for a given neural network architecture? 2. Given such an architecture, which training algorithms lead the model to fall into each regime?
> Answering these questions requires a deeper theoretical analysis of the properties of transformers, which is currently an open and challenging direction. While we do not yet have a definitive theoretical answer, our experimental results provide some insights. We will add a discussion of this point in the final section and consider it an important direction for future work.
>
> >Q2. The theory contribution is unclear. If my understanding is correct, the theory mainly states that there are two possible scenarios and does not directly relate to other parts of the paper, such as algorithms. The insight drawn from the theory is vague.
>
> A. Our theoretical results are directly connected to the algorithmic design and other components of the paper. Let us clarify. In Theorem 4.6, we prove that ECE ≤ TCE. In other words, obtaining a well-calibrated model reduces to finding one that is close to the **target probabilistic model**, i.e., the solution to the following constrained optimization problem:
>
> $$
> \max \text{ACC}(\pi) \quad \text{s.t.} \quad \text{ECE}(\pi) = 0.
> $$
>
> In Section 5, we discuss how to approximate ACC and ECE in the context of LLMs—specifically in Sections 5.1 and 5.2, respectively. Having defined approximations for the ACC and ECE losses, we introduce an EM algorithm to learn the underlying probabilistic model. This choice is motivated by our theoretical formulation of calibration as a probabilistic inference problem, for which EM is a natural and widely used solution.
>
> >Q3. There is only one baseline (temp scaling); it would be better to compare  with other baselines such as label smoothing, [1], etc.
>
> A. Thank you for the question. First, we would like to emphasize that temperature scaling is a strong baseline for reducing ECE. Prior work [3,4] has shown that it is generally more effective than label smoothing and other techniques for improving calibration.
> Nonetheless, we conducted additional experiments and found that, similar to image classification tasks, label smoothing remains less effective than temperature scaling on LLMs as well. We will add the experiments to our revised paper.
> | Model            | conf-ECE   |            | cw-ECE    |            | Accuracy  |            |
> |-|-|------------|-|-|-|-|
> |                  | In-Domain | Out-Domain | In | Out | In | Out |
> | Llama3.1-8B-Tulu | 0.1898     | 0.1009     | 0.0692    | 0.0639     | 0.6372    | 0.7116     |
> | Vicuna-7B        | 0.1221     | 0.0823     | 0.0517    | 0.0544     | 0.4517    | 0.5767     |
> | Olmo2-7B         | 0.1010     | 0.0499     | 0.0791    | 0.1298     | 0.6808    | 0.6431     |
> | Mistral-7B       | 0.1874     | 0.1121     | 0.0900    | 0.0990     | 0.6479    | 0.6997     |
>
>
>
> Regarding the contextual calibration method introduced in [1], we have carefully reviewed the paper. Although the term "calibration" appears in its name, the method is primarily designed to improve task performance across various NLP tasks, rather than to reduce ECE specifically. As such, their objective differs from ours, and the approaches are not directly comparable.
>
> [3] When Does Label Smoothing Help?
> Rafael Müller, Simon Kornblith, Geoffrey Hinton
>
> [4] On Calibration of Modern Neural Networks
> Chuan Guo, Geoff Pleiss, Yu Sun, Kilian Q. Weinberger
>
> >Q4. A better comparison of the Win rate is between CFT and DPO-generated responses since response qualities may decrease but still outperform the one being compared in the current setting.
>
> A: Thank you for the suggestion. We agree that comparing the win rate between CFT and DPO-generated responses provides more information. We have conducted these additional experiments on AlpacaEval dataset, and the results are now included in our revised paper and provided below for reference.
> |Win Rate | CFT| DPO | RCFT| DPO|
> |-|-|-|-|-|
> |  Llama-3.1-8B-Tulu | 51.68 | 48.32    | 46.83 | 53.16    |
> | Vicuna-7B | 46.46 | 53.54    | 50.43 | 49.57    |
> | Olmo2-7B | 62.48 | 37.52    | 46.12 | 53.88    |
> | Mistral-7B  | 46.96 | 53.04    | 49.81 | 50.19    |
>
> Experiments on Alpaca-Eval and Arena-hard are provided in the end of our response to Reviewer USq4.
>
> >Q5. Some relevant methods are not discussed: [1,2].
>
> A. We have included both [1] and [2] in the revised version of the paper.

---

> > ### Comment · Reviewer_QE3c · 2025-04-08
> >
> > Thanks for the rebuttal. My concerns are mostly addressed. Quick questions for a table in Q4: Why are there two DPO columns with different numbers?

---

> > > ### Author Response · Authors · 2025-04-09
> > >
> > > Thank you very much for your response. To clarify briefly: the first column under DPO shows the win rate compared to CFT, while the second column shows the win rate compared to RCFT. We realized that the previous table layout may have been unclear, so we have slightly revised the structure to present the results more clearly.
> > >
> > > |Win Rate                    | CFT vs DPO | RCFT vs  DPO |
> > > |-|-|-|
> > > |  Llama-3.1-8B-Tulu                            | 51.68 vs 48.32    | 46.83 vs 53.16    |
> > > | Vicuna-7B                                   | 46.46 vs 53.54    | 50.43 vs 49.57    |
> > > | Olmo2-7B                                     | 62.48 vs 37.52    | 46.12 vs 53.88    |
> > > | Mistral-7B                                   | 46.96 vs 53.04    | 49.81 vs 50.19    |
> > >
> > > Here is a more detailed explanation. Following your suggestion that “a better comparison of the win rate is between CFT- and DPO-generated responses,” we conducted the following experiments. Using the AlpacaEval dataset, we generated responses with our CFT/RCFT models and the DPO model. We then used GPT-4 as a judge to determine which response was better in each case.
> > >
> > > For example, in the first row (using the Llama3.1 model), when comparing CFT and DPO responses, GPT-4 judged 51.68% of CFT responses to be better and 48.32% of DPO responses to be better. These percentages sum to 100%. Similarly, in the comparison between RCFT and DPO, GPT-4 preferred 46.83% of RCFT responses and 53.16% of DPO responses.
> > >
> > > These results suggest that the generation quality of CFT and RCFT is approximately on par with DPO. Importantly, this also indicates that incorporating CFT does not degrade the alignment quality achieved by DPO.

---

### Decision · Program_Chairs · 2025-05-01

**Decision:**

Accept (poster)

**Comment:**

This paper addresses the poor calibration of Large Language Models (LLMs) after preference alignment (e.g., RLHF/DPO). The authors propose two methods: calibration-aware Fine-Tuning (CFT) and Regularized CFT (RCFT). CFT restores calibration without sacrificing accuracy, while RCFT balances calibration and accuracy for models in "non-calibratable" regimes. Theoretical analysis distinguishes calibratable/non-calibratable regimes based on accuracy thresholds.

Strengths:
1. The paper tackles a critical issue in LLM alignment.
2. Methodology: The EM-based CFT/RCFT is well-motivated, and experiments show consistent improvements across models and domains.

Concerns raised by the reviewers that haven't been fully addressed:
1. Theoretical Nuance: While bounds on TCE are proven, the practical distinction between calibratable/non-calibratable regimes (e.g., architecture-dependent thresholds) remains under-explored.
2. Evaluation Scope: Limited ablation studies (e.g., sensitivity to regularization parameter λ).
3. Generalization to a broader case (e.g. beyond multichoice problems of 4 options)

Note that the authors also address several questions raised by the reviewers regarding: baseline comparison on temperature scaling as well as presentations of the paper for better clarity.

Overall the paper is theoretically-grounded paper addressing important issues of miscalibration in LLM alignment post-training.  While minor issues persist, the rebuttal effectively addresses major concerns, and the core claims are validated.